# Innovative design of minimal invasive biodegradable poly(glycerol-dodecanoate) nucleus pulposus scaffold with function regeneration

Lizhen Wang[1,4], Kaixiang Jin [1,4], Nan Li[2], Peng Xu[1], Hao Yuan[1], Harsha Ramaraju[3], Scott J. Hollister[3] & Yubo Fan [1]✉

Minimally invasive biodegradable implants with regeneration have been a frontier trend in clinic. Degeneration of nucleus pulposus (NP) is irreversible in most of spine diseases, and traditional spinal fusion or discectomy usually injure adjacent segments. Here, an innovative minimally invasive biodegradable NP scaffold with function regeneration inspired by cucumber tendril is developed using shape memory polymer poly(glycerol-dodecanoate) (PGD), whose mechanical property is controlled to the similar with human NP by adjusting synthetic parameters. The chemokine stromal cell-derived factor-1α (SDF-1α) is immobilized to the scaffold recruiting autologous stem cells from peripheral tissue, which has better ability of maintaining disc height, recruiting autologous stem cells, and inducing regeneration of NP in vivo compared to PGD without chemokine group and hydrogel groups significantly. It provides an innovative way to design minimally invasive implants with biodegradation and functional recovery, especially for irreversible tissue injury, including NP, cartilage and so on.

Aging, injury, and other stimuli usually cause intervertebral disc degeneration, and more young patients suffered from the disease recently[1]. In disc degeneration, NP inside the disc occurs fibrosis, loses of hydration, and decreased support ability, leading to a reduction of disc height and herniation[2]. The NP cannot regenerate itself after degeneration nor maintain normal function when its surrounding tissue is damaged, including annulus fibrosus (AF) and cartilage endplate (EP)[3,4]. For nerve compression pain caused by disc height loss, discectomy and subsequent spinal fusion are suitable treatments in clinic[5,6]. However, the invasive intervention will inevitably damage the surrounding tissue[7]. In addition, the fused disc will also induce

adjacent segment disease for the limited mobility of disc[8]. Thus, minimally invasive functional regeneration of NP is a new trend in clinic[9].

With the rapid development of tissue engineering, stem cell therapy is used to differentiate NP cells and achieve NP reconstructions in previous studies[10,11]. It is proved that stem cell therapy slowed down or reversed NP degeneration[12–14]. But stem cell therapy also faces some problems, such as limited sources of autologous stem cells, long preparation periods, and immune rejection of allogeneic cells[15]. Noncellular therapies are proposed to solve the problems mentioned above, including bioactive factors, gene delivery, and extracellular

[1]Key Laboratory of Biomechanics and Mechanobiology (Beihang University), Ministry of Education, Beijing Advanced Innovation Center for Biomedical Engineering, School of Biological Science and Medical Engineering, School of Engineering Medicine, Beihang University, Beijing 100083, China. [2]Department of Spine Surgery, Beijing Jishuitan Hospital, The Fourth Clinical Medical College of Peking University, Beijing 100035, China. [3]Wallace H. Coulter Department of Biomedical Engineering, Georgia Institute of Technology and Emory University, 313 Ferst Drive, Atlanta, GA 30332, USA. [4]These authors contributed equally: Lizhen Wang, Kaixiang Jin. ✉e-mail: yubofan@buaa.edu.cn

vesicles[16–18]. A clue was found that the chemokine SDF-1α loaded in the hydrogel would induce the migration of autologous stem cells from surrounding tissue and peripheral blood into NP after implantation but failed in the long-term treatment[19–21]. It is thought that hydrogel scaffolds cannot provide sufficient mechanical support as strong as native NP, then the delivered ingredients are squeezed out along the surgical wounds in AF under compression, resulting in treatment failure in long-term[22,23]. Combined NP filling and AF repair is a feasible approach to solve the problem as previous mentioned[24]. A NP scaffold with mechanical support similar to native NP at restricted condition and minimal invasive ability is another potential approach we believe. It is necessary to develop an excellent degradable scaffold loaded SDF-1α with well-matched disc-supporting for tissue-engineered NP reconstruction[15,25]. The ideal tissue-engineered NP scaffold should meet characteristics including minimally invasive, disc-supporting, and viscoelastic properties matching with native NP[26,27], and regeneration with biodegradation of scaffold. PGD is a typical semi-crystalline biodegradable shape memory polymer for the continuously adjustable glass transition temperature, which is polymerized by glycerol and dodecanedioic acid[28–31]. The mechanical properties of PGD are close to the native soft tissue at body temperature[32,33]. So, it is supposed that PGD will be a good scaffold loading SDF-1α to the NP regeneration by adjusting its synthetic parameters.

In this study, an NP scaffold with regeneration is inspired by tendril of cucumber vine, which can be implanted to the nucleus cavity minimally in surgery (Fig. 1a, b). The scaffold is fabricated with PGD immobilized chemokine SDF-1α through a specific heparin linkage. The characteristics of the scaffold, including shape memory, mechanical support, and degradation rate, are adjusted and optimize to match native NP by synthetic parameters of PGD. Specifically, a PGD rod with small diameters at room temperature is delivered through the hollow needle into the nucleus cavity of New Zealand white rabbits (Fig. 1c). It will curl and deform to a vortex-plate shape with a bigger cross-section triggered by body temperature after implanted, which avoids it extrude from the punctured hole in AF under the condition of compression. It is found that the PGD NP scaffold can maintain the disc height, also recruit autologous stem cells, and regenerate degenerated NP through its released SDF-1α within 16 weeks. The design of a minimally invasive NP scaffold with regeneration inspired by a tendril of cucumber vine will be an innovative idea for treatment of disc degeneration, which also has potential application in clinic.

## Results

### Tailoring shape memory and mechanical properties of PGD
PGD samples with different shape memory and mechanical properties were synthesized using various synthetic parameters, including cure time (t) from 0-216 h and the molar ratio of hydroxyl to carboxyl in reactants (MR$_{H/C}$) from 0.75-2.00. And the thermodynamic properties of samples were analyzed using a differential scanning calorimeter (DSC). It was shown that PGD was a typical semi-crystalline polymer and its shape memory properties resulted from the formation and melt of internal crystals with temperature changes (Supplementary Fig. 1). The relationship between the melting peak as the transition temperature (T$_{trans}$) and synthetic parameters of PGD was obtained in Fig. 2a. T$_{trans}$ of PGD decreased linearly with the increase of cure time. The decrease rate was affected by MR$_{H/C}$ as described in Fig. 2a. The shape recovery rate (R$_r$) and fixation rate (R$_f$) of PGD samples at 37 °C and 20 °C were shown in Fig. 2b. R$_r$ and R$_f$ were all >98% when T$_{trans}$ of PGD was 27.8 °C - 36.6 °C. It would maintain the programmed shape at room temperature and restore the original shape at body temperature.

Twelve PGD material groups were selected according to different synthetic parameters with four MR$_{H/C}$ and three cure times in the range of optimized T$_{trans}$, as listed in Table 1. Young's modulus of twelve PGD samples were measured as shown in Fig. 2c. Young's modulus of the NP

scaffolds was calculated by numerical simulation based on mechanical properties of twelve different PGD material as shown in Fig. 2d. Then, a set of PGD synthesis parameters (MR$_{H/C}$ = 1.50, $t$ = 72 h) closest to the compressive modulus of native NP (1.01 ± 0.43 MPa) was finally selected and verified to fabricate our NP scaffold, also the compressive modulus of AF was known as 440–750 kPa[34]. As shown in Fig. 2e, the ex vivo disc implanted with PGD NP scaffold had a stress relaxation curve similar to that of the native intact disc, while the ex vivo discs without NP had lower axial stress after compression. As shown in Fig. 2f and Supplementary Movie 1 and Movie 2, the PGD NP scaffold passed through the needle and deformed to the vortex-plate shape like a cucumber tendril at 37 °C, which achieved invasive implantation minimally.

The numerical models of L5-L6 vertebra disc with PGD NP scaffold, hyaluronic acid (HA) NP scaffold, and native intact disc were developed based on Micro-CT images of the rabbit lumbar segment (Fig. 3a–e). Figure 3f showed Mises stress, pressure, and displacement of NP and AF under the condition of axial loading. For the native intact disc model, high-stress areas occurred in the anterior inner edge of AF. For the discs implanted with PGD or HA NP scaffold models, a high-stress area appeared in the posterior part of AF. Meanwhile, stress concentration occurred in the punctured hole. As shown in Fig. 3g, the maximum Mises stress on the PGD NP scaffold model was 12.9% lower than that of the HA NP scaffold model but 8% higher than that of the native intact disc model. Maximum pressures in NP under compression were 0.22, 0.19, and 0.09 MPa in the native intact disc, PGD NP scaffold, and HA NP scaffold model, respectively, as shown in Fig. 3h. The deformation of NP and AF after compression was shown in Fig. 3i and Supplementary Movie 3, 4. HA NP scaffold in the nucleus cavity would be squeezed out along the punctured hole, but the PGD NP scaffold did not.

### Biocompatibility, biodegradation, and chemokine immobilization of PGD NP scaffold
As shown in Fig. 4a, b, there was no significant difference in cell proliferation between the PGD and polycaprolactone (PCL) in the 7-days experiment. Figure 4c showed the PGD sample degraded to 94.80%, 47.47%, and 9.73% of its initial mass at 1, 4, and 9 weeks after subcutaneous implantation, and the relationship between mass loss and degradation time showed a linearly decreasing trend. There was no inflammation nor necrotic tissue in the peripheral tissue of the scaffold. Surface erosion was confirmed after comparing surface and cross-section views of scanning electron microscope (SEM) images from the PGD sample at different timepoint (Fig. 4d). 1000 ng recombinant human SDF-1α was immobilized on PGD NP scaffolds by specific heparin-mediated interaction (Fig. 4e). The chemokine dose was calculated based on the previously optimized dose for rats' disc and ten times multiplied according to the difference in body weight between rats and rabbits[20]. SDF-1α immobilized to the heparin-mediated PGD NP scaffold with a loading efficiency of 98.58% compared to 13.85% of the traditional physical method (Fig. 4e, f). In vitro SDF-1α release experiment proved that PGD NP scaffolds treated with the heparin-mediated method own slower release kinetics than that of the physical-adsorption treated hydrogels (Fig. 4g). PGD NP scaffolds released only one-third of its loaded SDF-1α in the first 12 h, while HA NP scaffolds had released over 50% simultaneously.

### Histological analysis based on in vivo implantation in rabbits' L5-L6 discs
As shown in Fig. 5a and Supplementary Fig. 2a, HE and immunofluorescence-stained images along the coronal axis of L5-L6 discs from the surgically treated rabbits at 8, and 16 weeks were sectioned, and microvascular were observed near EP as indicated by black/white arrows. CD90 & CD166 co-positive cells were highly expressed in the microvascular and bone marrow of EP. As shown in

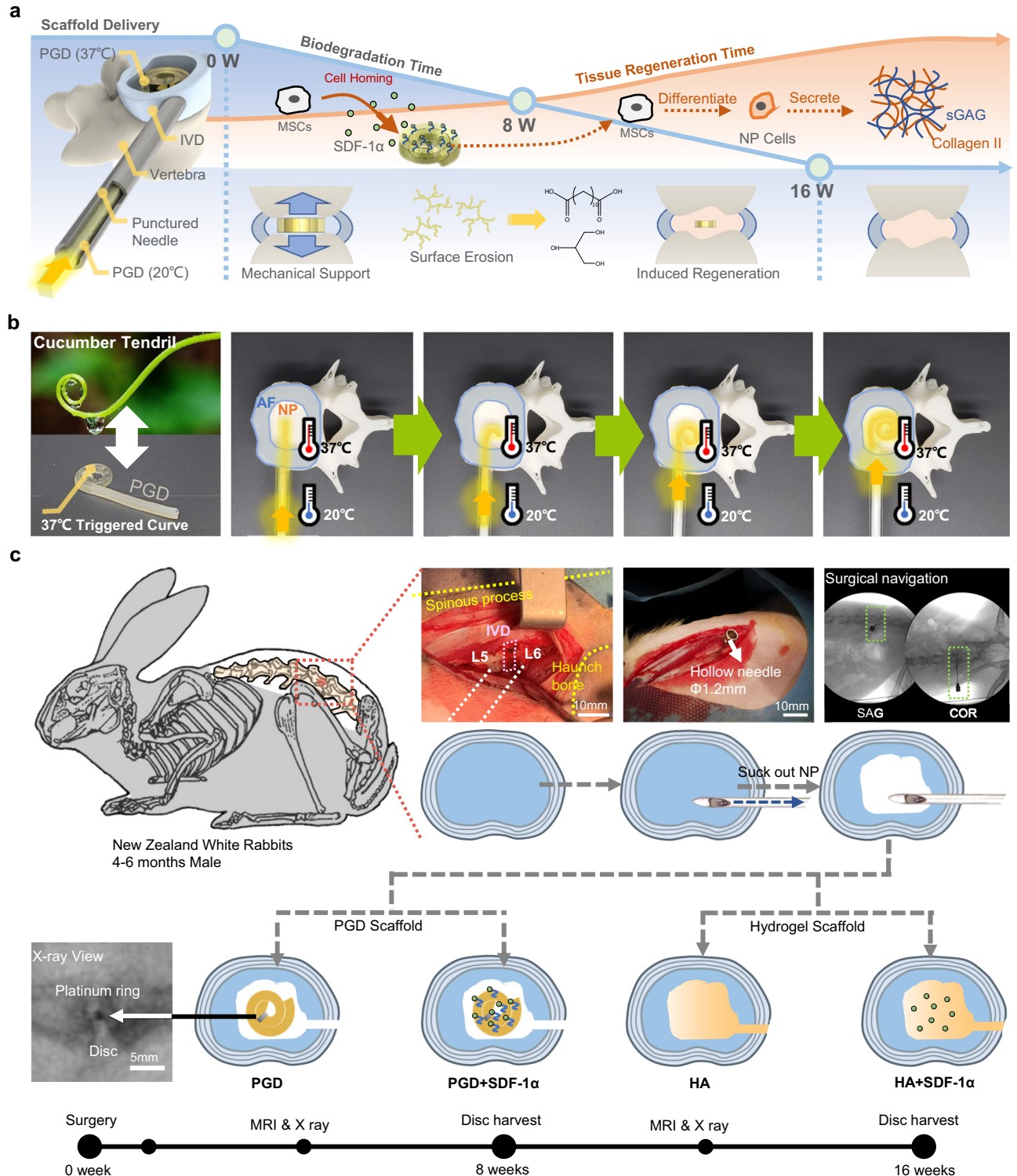

**Fig. 1 | Design and surgical procedure of biodegradable minimally invasive PGD NP scaffold with regeneration. a**, Schematic of the PGD NP scaffold's design. Disc height is maintained by the scaffold's mechanical support after implantation. SDF-1α immobilized on scaffolds will be released to recruit autologous MSCs from peripheral tissues or blood vessels. Extracellular matrix of NP will be regenerated gradually, and disc function will be recovered with the degradation of PGD scaffolds. **b**, Cucumber tendril inspired shape deformation design for invasive deliver minimally. PGD NP scaffold is minimally implanted into nucleus cavity using punctured needle, which deforms to vortex shape triggered by body temperature. **c**, Surgical procedure of NP scaffolds implanted in rabbits' L5-L6 disc. New Zealand white rabbits are selected and a hollow needle with 1.2 mm diameter is inserted into the nucleus cavity of the L5-L6 disc with the assistance of C-arm machine. A platinum ring is fixed on the NP scaffold to observe scaffold position through X-ray during surgery. Four NP scaffold groups are divided to implant PGD, PGD + SDF-1α, HA, HA + SDF-1α scaffolds respectively.

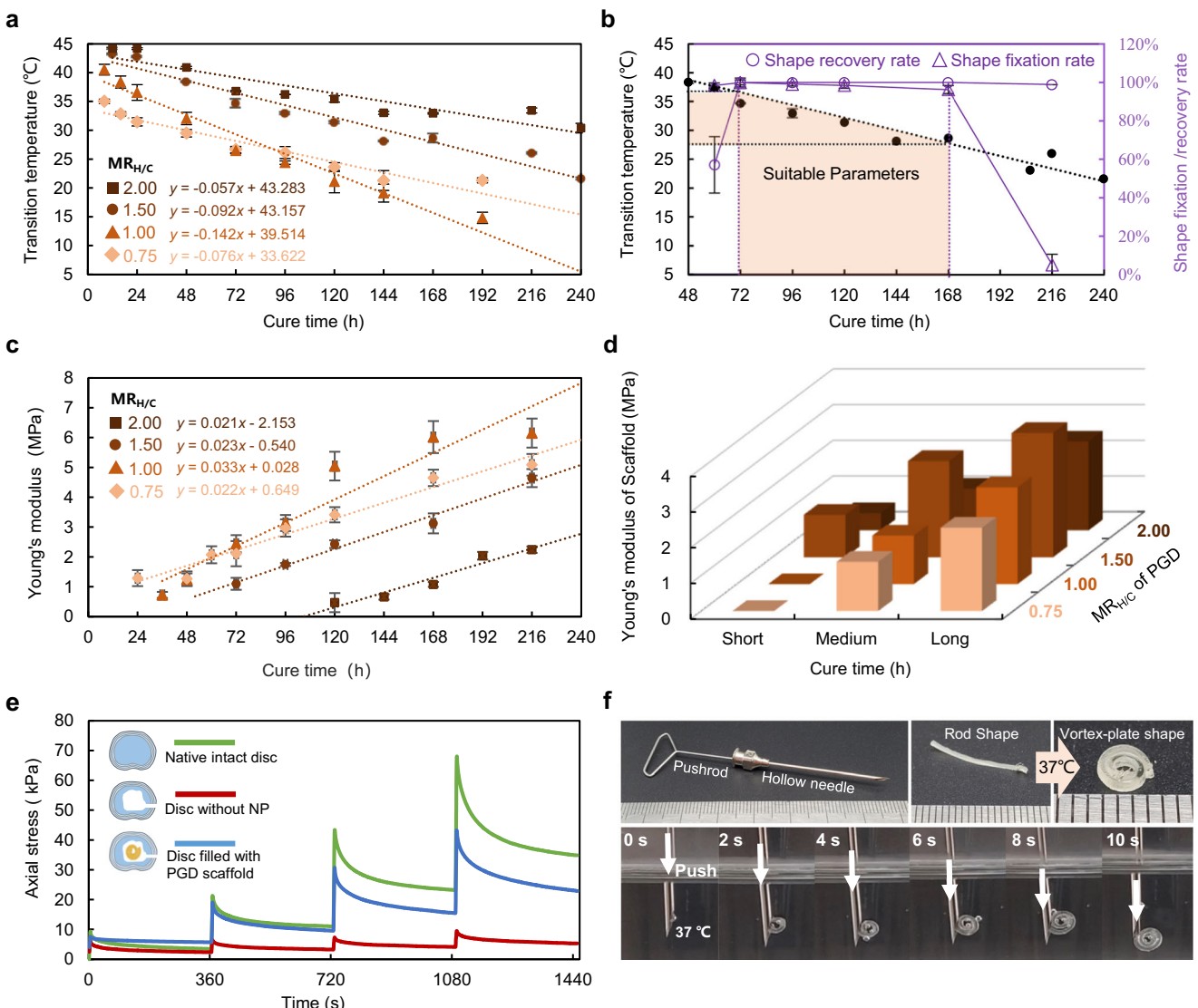

Fig. 2 | **Shape memory and mechanical properties of PGD by adjusting synthesis parameters. a** Transition temperatures of PGD varied with different synthesis parameters ($n = 5$) are presented as mean values ± SD. The linear functions between transmission temperature ($T_{trans}$) and cure time are showed at bottom left. **b** Shape recover rate ($R_r$ at 37 °C, $n = 5$) and shape fixation rate ($R_f$ at 20 °C, $n = 5$) of PGD with varied synthesis parameters are presented as mean values ± SD. $R_r$ and $R_f$ are >98% when $T_{trans}$ of PGD is 27.8 °C - 36.6 °C. **c** Young's modulus of PGD with varied synthesis parameters ($n = 6$) are presented as mean values ± SD, which increase linearly with cure time. **d** Young's modulus of PGD scaffolds with twelve synthetic parameters calculated based on numerical simulation. Long, Medium, Short represented the cure time when the transition temperature of PGD is -28 °C, -32 °C, and -36 °C respectively. **e** Four-step stress relaxation property of ex vivo lumbar discs of rabbits. The green line represents data from native intact disc, the red line represents disc without NP, and the blue line represents disc filled with PGD scaffold. **f** Shape deformation process of PGD NP scaffold made by the optimized synthetic parameter ($MR_{H/C} = 1.50$, $t = 72$ h). The PGD NP scaffold is programmed and maintain the rod-like shape at room temperature, which can be inserted into a hollow needle and curled like a cucumber tendril at 37 °C.

Table 1 | **Young's modulus of PGD raw material, FEA simulated PGD scaffolds, and actual PGD scaffolds made by varying $MR_{H/C}$ and cure time**

| $T_{trans}$ (°C) | | Young's Modulus at ~37 °C (MPa) | | | | | | | | Actual Scaffold (Experiment) |
|---|---|---|---|---|---|---|---|---|---|---|
| | Cure Time (h) | PGD | | | | Simulated Scaffold (FEA) | | | | |
| | | $MR_{H/C}$ | | | | $MR_{H/C}$ | | | | $MR_{H/C}$ |
| | | 0.75 | 1.00 | 1.50 | 2.00 | 0.75 | 1.00 | 1.50 | 2.00 | 1.50 |
| ~28 | Long (48-216 h) | 2.11 ± 0.42 | 2.44 ± 0.29 | 3.13 ± 0.34 | 2.25 ± 0.11 | 2.34 | 2.71 | 3.48 | 2.49 | Fragile |
| ~32 | Med (24-168 h) | 1.29 ± 0.27 | 1.24 ± 0.20 | 2.43 ± 0.13 | 1.08 ± 0.08 | 1.38 | 1.36 | 2.70 | 1.17 | 2.49 ± 0.23 |
| ~36 | Short (24-120 h) | N/A | Uncured | 1.10 ± 0.20 | 0.47 ± 0.32 | Uncured | Uncured | 1.19 | 0.48 | 1.23 ± 0.11 |

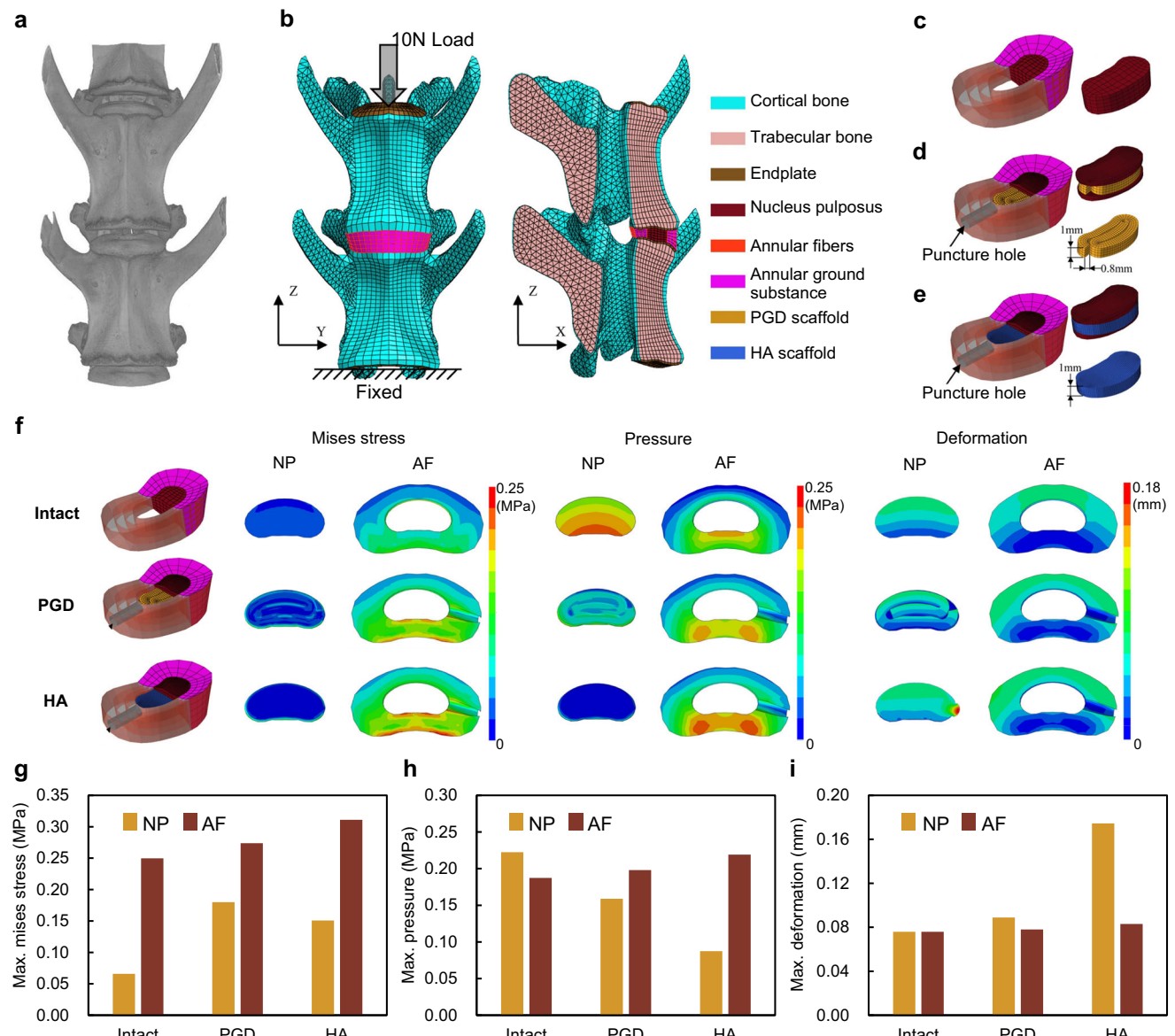

**Fig. 3 | Finite element analysis of rabbit L5-L6 lumbar disc with native NP, PGD NP and HA NP. a** Anatomical structure of rabbit's L5-L6 lumbar vertebral based on Micro-CT images. **b** Boundary conditions and meshed elements of L5-L6 disc model. **c**–**e** Finite element models of native intact disc (**c**), disc implanted with PGD NP scaffold (**d**), and disc implanted with HA NP scaffold (**e**). **f** Mises stress, pressure, and deformation of NP and AF in the above mention models under compression. **g**–**i** The comparison of maximum Mises stress (**g**), pressure (**h**), and deformation (**i**) of NP and AF under compression.

Fig. 5b and Supplementary Fig. 2, Cells in NP of intact group were stellar shape surrounded with dense eosin-stained extracellular matrix; Cell numbers in NP of NP Injury group was low, and most of their morphology was small and flat without clear cytoplasm at both 8 weeks and 16 weeks after implantation; Cells in NP of PGD group were small and flat, which reflected a degenerated morphology after 8 weeks implantation; Cells in NP of PGD + SDF-1α group were chondrocyte-like morphology with rounded nucleus, clear cytoplasm and eosin stained extracellular matrix, and most of the cells with these features were CD90 & CD166 co-positive cells; Cells in NP of HA and HA + SDF-1α groups were clustered together with both flatted and rounded shape, which showed a typical apoptotic morphology as previously mentioned[35,36]. The number of microvascular in each group at 8 weeks was counted as shown in Supplementary Data 1 and plotted in Fig. 5c, and all the groups were significantly different from the native intact disc. PGD + SDF-1α group was 86.33% higher than the intact one and NP Injury disc ($P < 0.01$), while the other groups were significantly lower than the intact one ($P < 0.05$). As shown in Fig. 5d, the number of

microvascular showed no difference at 16 weeks between the three groups (PGD, PGD + SDF-1α, and HA + SDF-1α) and the intact one, respectively ($P > 0.05$); HA group were significantly lower than that of the native intact disc group ($P < 0.01$); NP injury group was significantly lower than that of the intact, PGD, PGD + SDF-1α, and HA + SDF-1α groups ($P < 0.05$). As shown in Fig. 5e, f and Supplementary Data 2, the cells stained by CD90 and CD166 were counted and normalized with total cells in NP. The percentage of CD90 & CD166 co-positive cells of the PGD + SDF-1α group at 8 weeks was highest among all of the experimental groups, intact group and NP injury group ($P < 0.05$), while it was lower for the NP injury, PGD and HA groups than the intact one ($P < 0.05$). As shown in Fig. 5f, the percentage of CD90 & CD166 co-positive cells at 16 weeks had no difference between the three groups (PGD, PGD + SDF-1α, and HA + SDF-1α) and the intact one, respectively. For the NP injury and HA group, it was significantly lower than the other three groups (Intact, PGD + SDF-1α, and HA + SDF-1α) ($P < 0.05$).

As shown in Fig. 6a, the intact disc was found to have a clear NP-AF boundary and well-organized collagen lamellae without ruptured or

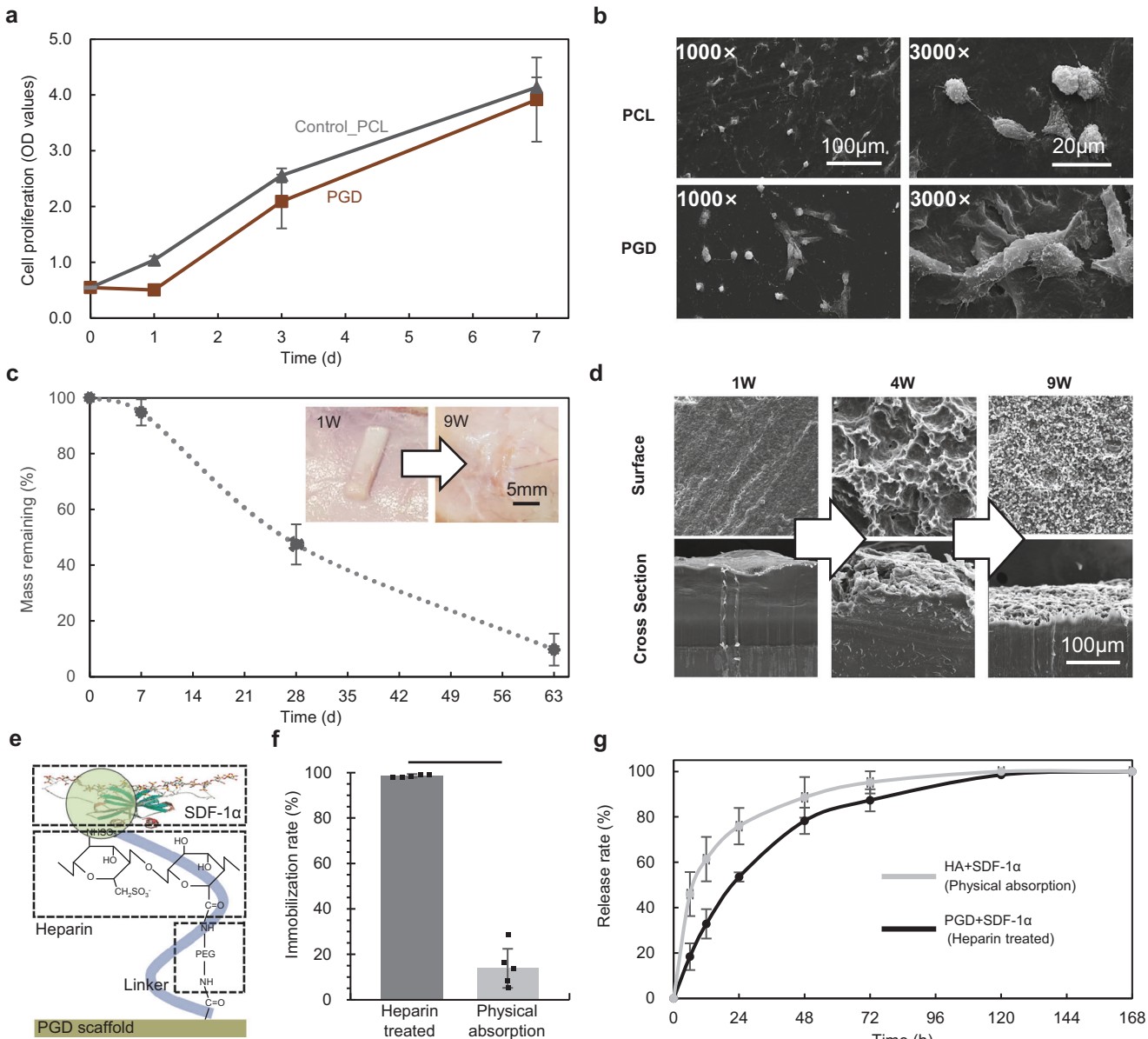

**Fig. 4 | In vitro cell proliferation, in vivo subcutaneous degradation, and chemokine SDF-1α immobilization of PGD NP scaffold. a** Cell proliferation of PGD samples ($MR_{H/C}$ = 1.50 and $t$ = 72 h, $n$ = 5) compared with PCL samples as control group, the data are presented as mean values ± SD. **b** Cell morphology on the surface of PGD and PCL samples are observed by SEM at 3 days. **c** Mass remaining of PGD samples ($n$ = 4) in subcutaneous degradation of Sprague-Dawley rats, the data are presented as mean values ± SD. Morphology of PGD samples and

its peripheral tissues at 1 and 9 weeks after implantation is shown at the upper right. **d**, Micro-morphology of the surface and cross-section of PGD at 1, 4, and 9 weeks after implantation. **e** Chemokine SDF-1α is immobilized on heparinized PGD scaffolds using specific heparin-mediated interaction. **f, g** In vitro evaluation of immobilized efficiency (**f**)($n$ = 5) and release (**g**) ($n$ = 5) of SDF-1α in PGD and hydrogel scaffolds respectively, the data are presented as mean values ± SD.

serpentine fibers, and no fibrotic tissue was observed in NP from the Masson-stained images, blue proteoglycan was rich in surrounding cells in NP which were homogeneously distributed as shown in Alcian Blue-stained images; For the NP Injury group, AF maintained its concentric circles morphology as intact disc but with serpentine fibers in it at 8 weeks, AF appeared inward bulging and the disc height decreased at 16 weeks, fibrosis in NP cavity and osteophyte in outer AF were observed at the same time; For the PGD group, a PGD NP scaffold was observed in the nucleus cavity, which means that it did not wholly degrade at 8 weeks, and it was similar to native intact disc in Masson and Alcian Blue staining. PGD NP scaffold was degraded entirely at 16 weeks, and prominent serpentine fibers were observed in AF. Fibrosis occurred in NP, and disc height decreased. No obvious proteoglycan was observed in NP; For the PGD + SDF-1α group, the PGD

NP scaffold was also not completely degraded at 8 weeks, and it showed slight degenerative features compared with the intact group, including serpentine fibers in AF and misty NP-AF boundary. And there was no obvious variation for the NP and AF at 16 weeks; For the HA group, inward bulging of AF was observed clearly from HE-stained images at 8 weeks, which led to disc height decreasing and the nucleus cavity compressed. AF inserted into the nucleus cavity was found, and more than one-third of AF bulged inward, obviously at 16 weeks. Collagen I was distributed in NP from Masson-stained images, which means severe fibrosis of NP, and osteophytes formed surrounding the punctured hole in AF; For the HA + SDF-1α group, inward bulging of AF was also observed from HE-stained images at 8 weeks. Collagen I was rich in NP based on Masson-stained images, and osteophytes also formed surrounding the punctured hole in AF, and Collagen I in NP

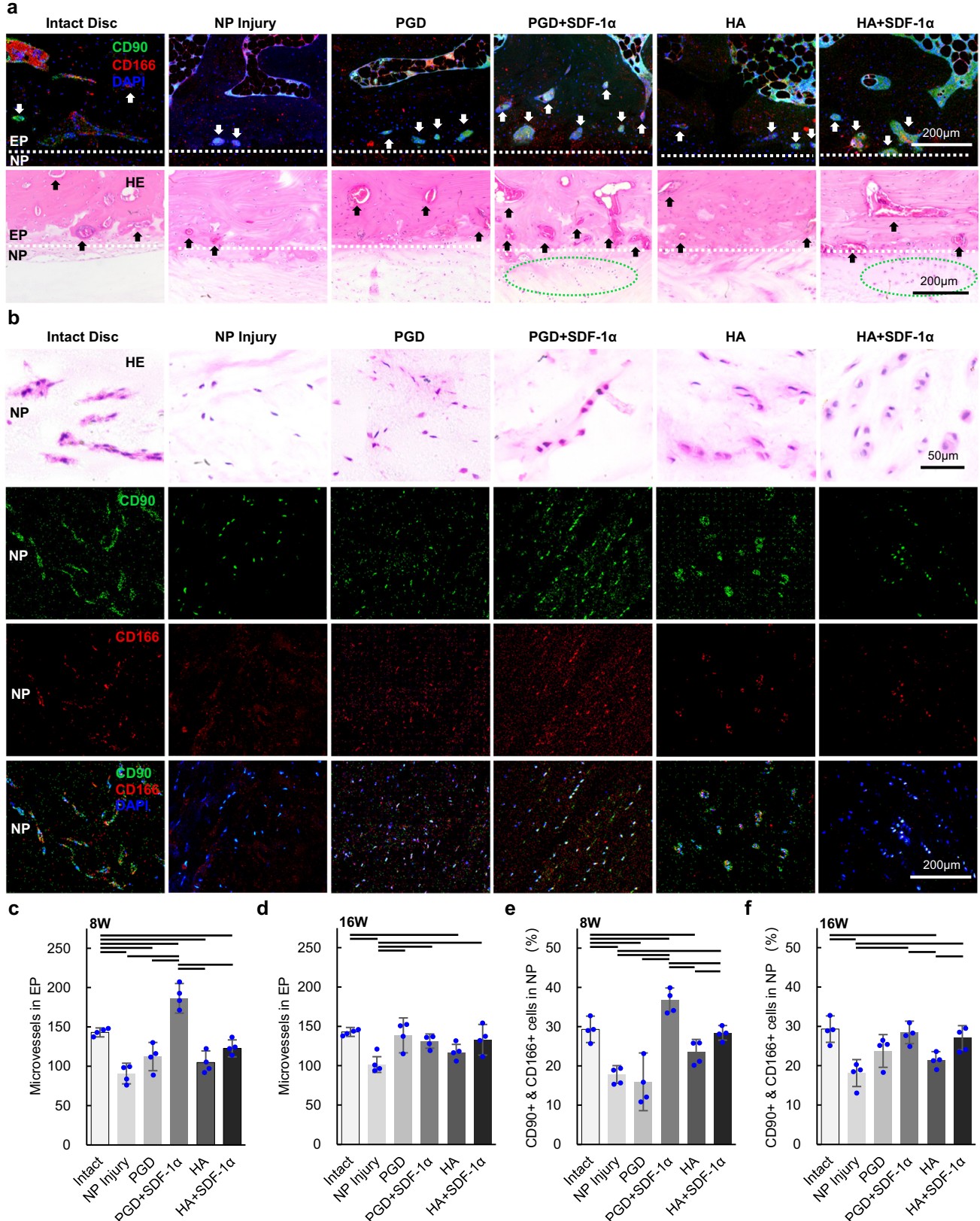

increased at 16 weeks, but proteoglycan was hardly found in NP. As shown in Supplementary Fig. 3, pathological sections stained with Safranine O and Masson through the puncture site were selected to clarify AF injury of each group at 8 and 16 weeks. The AF puncture injury of each group did not heal completely, and it was still clear at 16 weeks after implantation. Organization of AF in NP injury, HA, and

HA + SDF-1α groups showed typical degenerated characteristics. Thereinto, AF organization near the injury site in NP injury group showed inward convex into NP cavity due to the loss of NP in disc; hydrogel-injected HA and HA + SDF-1α groups showed both inward convex and outward convex near the injury site. Osteophytes were also formed outside the puncture injury, which was related to extrusion of

**Fig. 5 | Tissue morphology and cell composition in NP of Intact, NP Injury, PGD, PGD + SDF-1α, HA, and HA + SDF-1α group. a** HE and immunofluorescence-stained histological images of the boundary between NP and EP at 8 weeks after implantation. White dotted line represents the boundary between EP and NP, the black/white arrows indicate micro-vessels in EP, and the green dotted circle represents chondroid cells in NP. **b** HE and immunofluorescence-stained histological images of NP at 8 weeks after implantation. **c, d** Number of microvascular in EP at 8 weeks (**c**)($n = 4$) and 16 weeks (**d**)($n = 4$) implantation for native intact disc, NP injury disc, and above mentioned four scaffold groups. **e, f** Percentage of CD90 & CD166 copositive cells in NP at 8 weeks (**e**)($n = 4$) and 16 weeks (**f**)($n = 4$) for native intact disc, NP injury disc, and above mentioned four NP scaffold groups. The above data are all presented as mean values ± SD. Transverse lines indicate statistical differences between two groups analyzed by two-tailed $t$-test ($P < 0.05$), and the exact $P$-values are listed in the Supplementary Data 1 and 2.

the contents in NP after surgery. AF organization near the injury site in PGD and PGD + SDF-1α groups maintained well at 8 weeks, indicating sufficient mechanical support provided from the PGD NP scaffold. AF injury of PGD + SDF-1α group improved compared with the NP injury and other three scaffold groups at 16 weeks, and no abnormal morphology was observed during the period. As shown in Supplementary Fig. 4 and Supplementary Table 1, osteophytes incidence among each group was analyzed through HE-stained disc sections. The results showed that osteophytes incidence of the disc in NP Injury group, HA group and HA + SDF-1α group were about 75%. Osteophytes occurred in only 1 of the 8 discs in both PGD group and PGD + SDF-1α groups, among which osteophytes in the disc of PGD + SDF-1α group was milder than the other groups. The discs in NP Injury group, HA group and HA + SDF-1α group were all lack of mechanical support, which led to loss disc height and generated osteophytic after implantation. It was worth noting that osteophytes incidence in HA + SDF-1α group at 8 weeks was significantly higher than that in other groups, which might be related to its inappropriate stem cells recruited in the outer AF. One case of mild osteophyte in PGD + SDF-1α group might also be related to improper SDF-1α release at outer disc. Disc degeneration in each group was evaluated and quantified as histological scores according to the previously mentioned histological scale (Supplementary Table 2)[37]. As shown in Fig. 6b and Supplementary Table 3, the histological scores of each group were significantly higher than that of the native intact disc at 8 weeks ($P < 0.05$). Histological scores of the PGD + SDF-1α group were significantly lower than that of the HA group and HA + SDF-1α group, respectively ($P < 0.05$). At 16 weeks, histological scores of the PGD + SDF-1α group were close to the native intact disc ($P > 0.05$), while histological scores of the other groups were still higher significantly than those of the native intact disc ($P < 0.05$), as shown in Fig. 6c. The statistical differences among the groups were shown in Supplementary Data 3.

X-ray and T2 weighted MRI images of the native intact disc and four experimental groups at 1, 8, and 16 weeks after implantation were shown in Fig. 7a. Normalized Disc height index (DHI) was calculated as shown in Fig. 7b, d, f, h and Supplementary Data 4. DHI of PGD and PGD + SDF-1α groups at 1 week were 77.76% and 86.63% of their initial state, which was significantly >67.82%, 62.34% and 67.32% of HA, HA + SDF-1α and NP injury groups, respectively ($P < 0.05$). DHI of the PGD + SDF-1α group maintained above 80% in 1 ~ 16 weeks (Fig. 7h and Supplementary Table 4). There was no significant difference between the PGD group and PGD + SDF-1α group for DHI before 8 weeks ($P < 0.05$). After that, DHI of the PGD group gradually decreased from 77.91% at 8 weeks to 64.49% at 16 weeks (Fig. 7h). DHI of the HA and NP injury group decreased gradually in 1 ~ 16 weeks, and it was 53.79% and 53.39% of the native intact disc at 16 weeks, which was lower than that of all other times. DHI of the HA + SDF-1α group changed a little in 1 ~ 16 weeks and maintained at 71.20% of its initial state at 16 weeks after implantation. As shown in Fig. 7c and Supplementary Data 5, the T2 MRI signal intensity of NP in each experimental group significantly decreased at 1 week, and it decreased to about 60% of the preoperative state. But there was not variation for each experimental group until 8 weeks ($P > 0.05$). T2 MRI signal intensity of NP in PGD, HA and NP Injury groups decreased gradually from 48.32%, 57.41% and 47.38% at 8 weeks to 45.75%, 51.71% and 45.33% at 16 weeks separately (Fig. 7e, g). As shown in Fig. 7i and Supplementary Table 4, it increased from 66.43% at 8 weeks to 75.22% for the PGD + SDF-1α group at 16 weeks.

And it also increased from 64.46% at 8 weeks to 71.42% for the HA + SDF-1α group at 16 weeks. MRI images of the spine were used to analyze compressive myelopathy that might be caused by disc dislocation, spinal stenosis, or spinal fracture. Only one case of mild nerve compression was found in HA + SDF-1α group, and it did not cause hind limb paralysis of the rabbit. Severe compressive myelopathy causes pain and hind limb paralysis, and this phenomenon did not occur in all the experimental rabbits in this study. As shown in Fig. 7k, l and Supplementary Data 6, the effective instantaneous modulus ($E_{ins}$) and equilibrium modulus ($E_{equ}$) of the discs in each experimental group and NP injury group at 16 weeks after implantation were lower significantly than those of the native intact disc ($P < 0.05$). $E_{ins}$ and $E_{equ}$ of the PGD + SDF-1α group were 50.17 kPa and 37.24 kPa, which had no significant difference from the intact one ($P > 0.05$). $E_{ins}$ and $E_{equ}$ of NP injury, PGD, HA, and HA + SDF-1α were all significantly lower than those of the native intact disc ($P < 0.05$). $E_{equ}$ of HA and HA + SDF-1α group were 13.82 kPa and 9.20 kPa, which were significantly different compared from that of the PGD + SDF-1α group ($P < 0.05$).

## Discussion

A biodegradable minimal invasive NP scaffold with regeneration based on thermosensitive shape memory polymer PGD is proposed in this study, and its ability of degradation and regeneration is verified by a series of in vitro experiments and in vivo implantation of rabbit lumbar disc. Owning to the thermosensitive shape memory property of PGD, the scaffold possesses a cucumber tendril-like deformable structure that enables it to transform from a rod state at room temperature to a vortex-plate state after being implanted in the body. It is essential to design good mechanical properties of the scaffold to match with the native NP[15,25]. It is a benefit to delivery in surgery since PGD has a higher elastic modulus than elastoplastic at room temperature (Supplementary Fig. 5). The elastic modulus of PGD is close to the native NP at body temperature, which is a benefit to absorbing impact energy and maintain the disc height. The mechanical support of the PGD NP scaffold avoids squeezing out of the surgical hole after implantation compared with the HA NP scaffold group, as mentioned in previous studies[22–24]. Chemokine SDF-1α is immobilized on the PGD NP scaffold, and the initial mechanical support of the scaffold provides a better environment than the HA group in NP for tissue regeneration, then restores NP function, obviously.

Shape memory and mechanical properties of PGD were adjusted by varied synthetic parameters of polymer, including $MR_{H/C}$ and cure time. The optimized PGD synthetic parameters ($MR_{H/C} = 1.50$, $t = 72$ h) were selected to fabricate NP scaffold (Table 1), which had body temperature triggered deformation and matched mechanical property with native NP, achieving minimally invasive implantation and mechanical support in regeneration. The mechanical support and resilience characteristics of the PGD NP scaffold were close to the native intact disc according to the results of the stress relaxation experiment in ex vivo lumbar discs of rabbits[26,27] (Fig. 2e). Good biocompatibility of the PGD NP scaffold was also verified with in vitro cell proliferation and in vivo rats' subcutaneous implantation (Fig.4a, c). The mass loss decreased with the degradation of PGD linearly during the 9 weeks subcutaneous implantation. In vivo NP regeneration generally took over 8 weeks, indicating degradation duration of PGD NP scaffold met the requirement of the regenerate time in previous studies[12,17,20,38]. Surface erosion was observed in SEM images of PGD

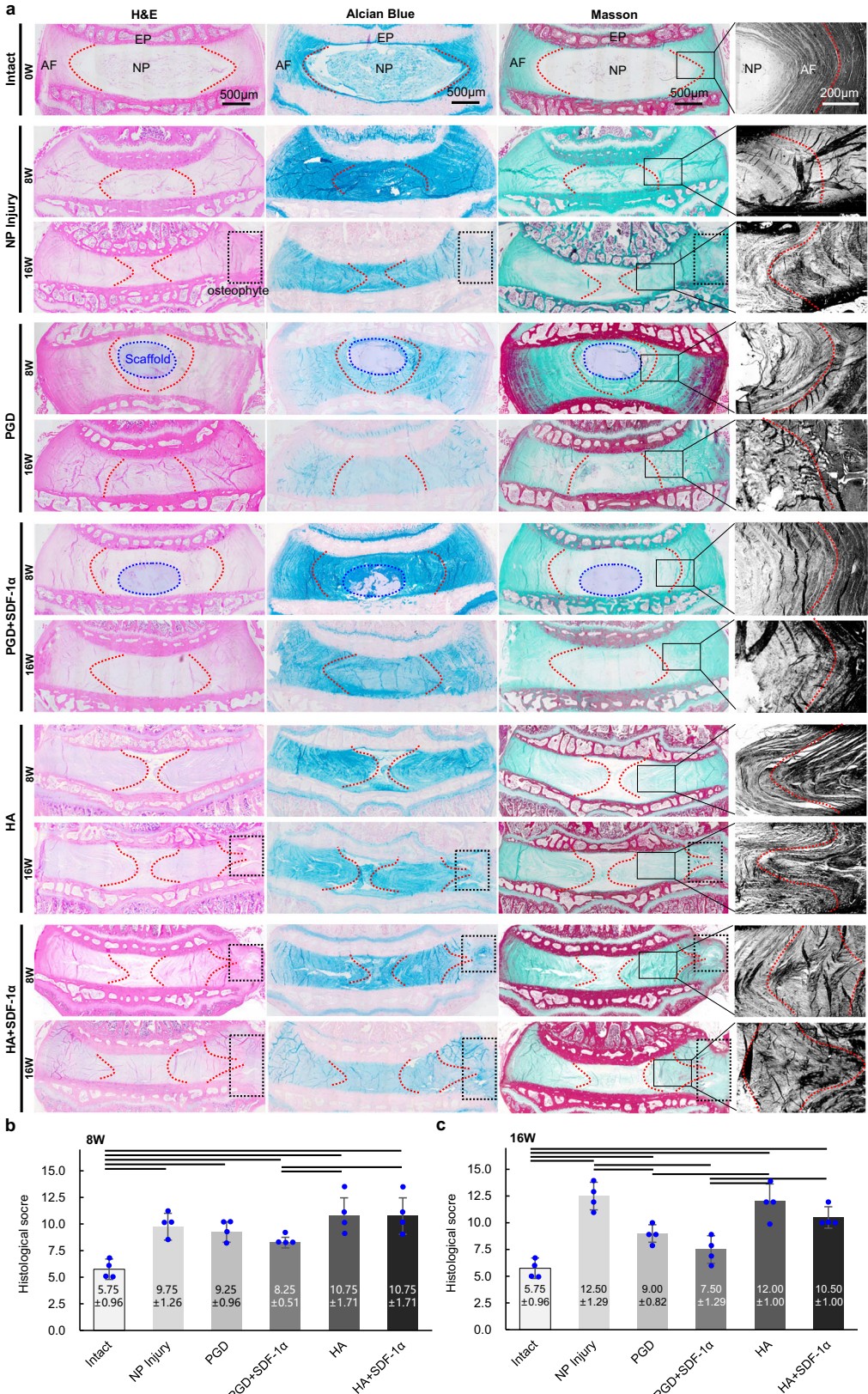

**Fig. 6 | Histological paraffin sections images and evaluation of disc degeneration after 8, 16 weeks implantation. a** HE, Masson and Alcian blue stained images of native intact disc, NP injury disc, and disc implanted with above mentioned four NP scaffolds in the coronal plane at 8 and 16 weeks. Blue dotted circle represents PGD scaffold in NP. Black dotted box indicates areas of osteophyte generated. Red dotted lines represent outline of AF in disc. **b, c** Histological scores of disc degeneration for native intact disc, NP injury disc and four experimental groups at 8 weeks (**b**)($n = 4$) and 16 weeks (**c**)($n = 4$), the data are presented as mean values ± SD. Transverse lines indicate statistical differences between the two groups determined by the two-tailed $t$-test ($P < 0.05$), and the exact $P$-values are listed in the Supplementary Data 3.

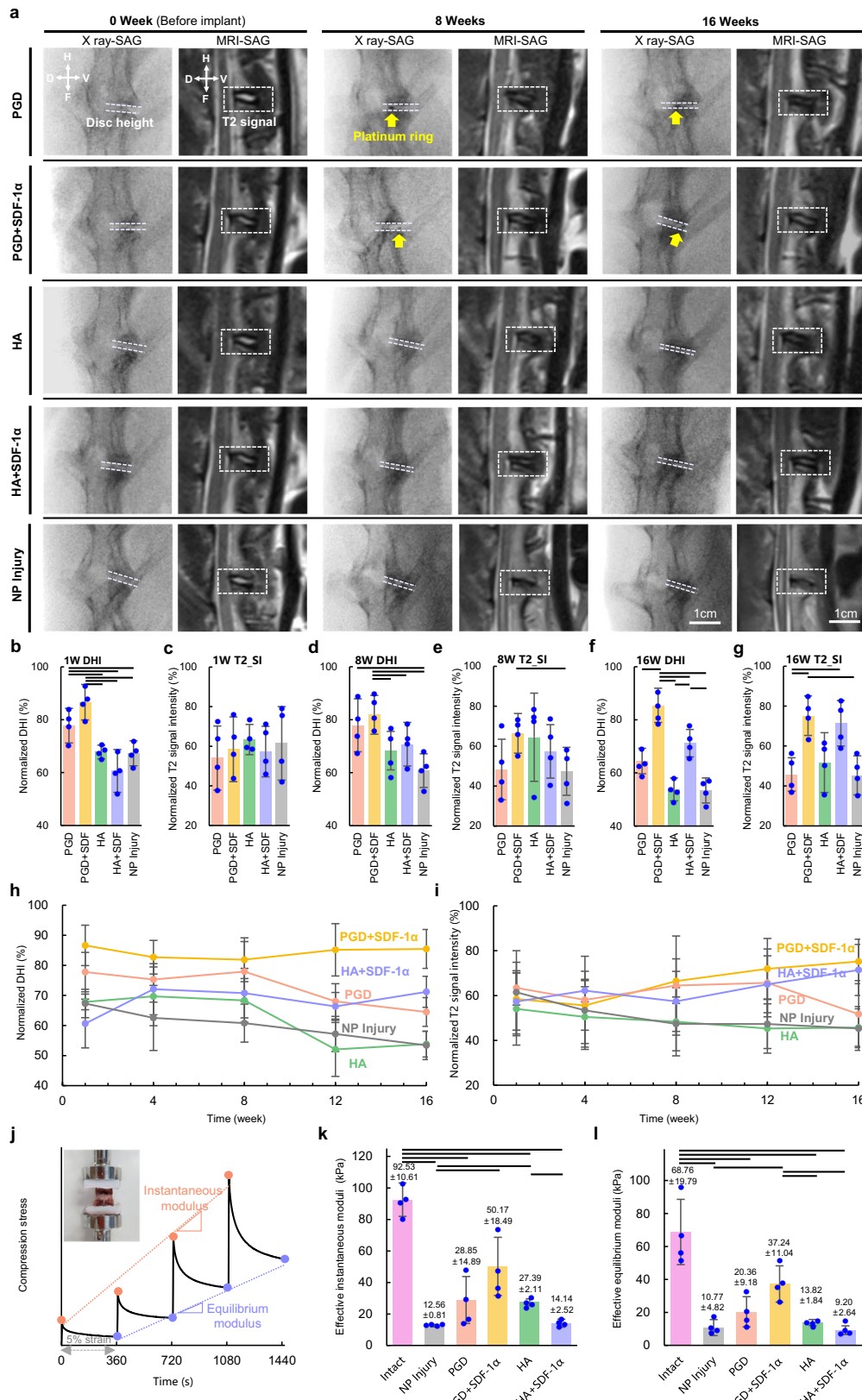

(Fig. 4d), which was also consistent with previous studies[32,39]. It was proved that optimized minimally invasive PGD NP scaffold in this study would provide mechanical support and enough degradation time to regeneration.

As we know, minimally invasive surgery has become a popular trend in clinic due to its small incision, slight pain, and quick postoperative recovery[40,41]. It relieves clinical symptoms and avoids irreversible damage to the peripheral tissue for degenerated NP[24,42]. The body temperature-triggered biomimetic deformable PGD NP scaffold was designed inspired by the cucumber tendril in this study, which would be implanted by minimally invasive surgery. It had a rod-like shape at room temperature and was delivered to the nucleus cavity

**Fig. 7 | Variation of disc height and T2 MRI signal intensity of NP during the 16 weeks implantation, and mechanical properties of the discs at 16 weeks.**
**a** X-ray and T2 MRI images of the discs at 0, 8, and 16 weeks (anatomical directions: H Head, F Foot, V Ventro, D Dorsal). The yellow arrow represents platinum ring on the PGD scaffold. **b**, **d**, and **f** Normalized DHI of the discs in NP injury disc and four experimental groups at 1 (**b**)($n = 4$), 8 (**d**)($n = 4$) and 16 weeks (**f**)($n = 4$) after implantation. **c**, **e**, and **g** Normalized T2 signal intensity of NP in the discs of NP injury and four experimental NP groups at 1 (**c**)($n = 4$), 8 (**e**)($n = 4$), and 16 weeks (**g**) ($n = 4$) after implantation. **h** Variation of normalized DHI of NP Injury and four experimental groups ($n = 4$) during 16 weeks. **i** Variation of normalized T2 signal

intensity in NP of NP injury disc and four experimental groups ($n = 4$) during 16 weeks. **j** Four-step compression stress relaxation for the collected L5-L6 discs after 16 weeks implantation. The discs are removed from peripheral tissue and fixed in the cured acrylate for mechanical experiments as shown in the upper left picture. **k**, **l** Effective instantaneous (**k**), and equilibrium moduli (**l**) are calculated based on stress-strain curves ($n = 4$). The above data are all presented as mean values ± SD. Transverse lines in figures indicate statistical differences between the two groups determined by the two-tailed $t$-test ($P < 0.05$), and the exact $P$-values are listed in the Supplementary Data 4, 5 and 6.

through a minimally invasive channel (Supplementary Movie 1). The PGD NP scaffold could automatically curl into a vortex-plate shape at the nucleus cavity from room temperature to body temperature. Then the deformed PGD NP scaffold achieved an increased cross-sectional area and matched with the mechanical properties of native NP after implantation. It did not squeeze out from the punctured hole nor dislocate in L5-L6 lumbar discs in the following 16 weeks in vivo experiments in the PGD and PGD + SDF-1α group, which avoided the failure of regeneration caused by lack of mechanical support and chemokine went out of NP and AF under compression. The platinum developing ring was fixed on a PGD NP scaffold to observe the visualization in implant surgery by C-arm machine (Fig. 1c). It was verified by in vivo delivery experiments with a hollow needle (1.2 mm diameter) into rabbits' L5-L6 discs, and the surgical operation was user-friendly and just took 10 min for the whole process.

Chemokine SDF-1α recruits cells with CXC chemokine receptor 4 such as bone marrow derived MSCs to participate in tissue regeneration[17,19,21,43,44]. Recent studies have shown that delivering SDF-1α in NP significantly promotes the migration of autologous MSCs and restores degenerated disc function in the non-stress load coccygeal disc model[20]. The mechanism of MSC promoting disc repair is still unclear, and current studies believe that tissue repair function of MSC is closely related to its secretion and extracellular vesicles mediated inhibitory effect of inflammation and apoptosis[9]. Mechanical environment has also been shown to affect differentiation effect of MSCs into NP cells through the YAP1 signaling pathway[12]. In this study, chemokine SDF-1α immobilized on PGD and HA NP scaffolds were used to achieve autologous MSCs recruitment and NP regeneration. Heparin-mediated specific interaction was used to immobilize SDF-1α on the PGD NP scaffold efficiently[43].

It was found that T2 MRI signal intensity of NP at 1 week after surgical treatment was used as the clinical indicator to reflect NP damage among each group (PGD, PGD + SDF-1α, HA, HA + SDF-1α, and NP Injury group) in this study, and they were all ~60% of their initial state before surgery with no statistical difference (Fig. 7c), which meant they had similar disc degeneration level after surgical treatment at the beginning. DHI of PGD and PGD + SDF-1α groups maintained 77% and 86% of their initial state separately, but HA and HA + SDF-1α groups were only 68% and 62% at 1 week (Fig. 7h), which indicated PGD NP scaffold had a better ability to support than HA NP scaffold. T2 MRI signal intensity of NP in PGD + SDF-1α and HA + SDF-1α groups increased with the increasing implantation time. But the signal intensity of PGD and HA groups decreased gradually (Fig. 7i). For the PGD + SDF-1α group and HA + SDF-1α group, DHI maintained the same in 1 ~ 16 weeks after implanted. For the PGD group, DHI kept the same in 1 ~ 8 weeks but decreased to 65% of the native intact disc at 16 weeks after implanted (Fig. 7h), which resulted from the biodegradation of the PGD NP scaffold at 8 ~ 16 weeks. For the HA and NP Injury group, DHI decreased gradually from 1 ~ 16 weeks after implanted, indicating their poor mechanical support[22–24]. PGD NP scaffold still existed in NP at 8 weeks and biodegraded entirely at 16 weeks based on the histological staining images (Fig. 6a). Fibrosis degeneration in NP occurred at 8 weeks in HA and HA + SDF-1α groups, but the same phenomenon did not occur in PGD group until 16 weeks based on the Masson-

stained images. But no fibrosis NP was observed in the PGD + SDF-1α group during the whole period. It meant that PGD could not maintain the disc height for its biodegradation from 8 ~ 16 weeks, and the scaffold-loaded chemokine SDF-1α played a vital role in NP regenerations. MSCs could be specifically stained by CD90, CD105, and CD166, etc. as previous mentioned[45]. CD90 and CD166 were selected as specific biomarkers in this study to characterize the MSCs proportion in NP. The MSCs proportion in NP of PGD + SDF-1α group at 8 weeks was 36.85%, which was significantly higher than that of intact group (29.32%), NP Injury group (17.84%), PGD group (15.96%), HA group (23.53%), and HA + SDF-1α group (28.43%). CD90 and CD166 co-positive cells were also highly expressed in the microvascular and bone marrow of EP, and the number of microvascular in PGD + SDF-1α group was significantly higher than that in PGD group and NP Injury group at 8 weeks. This phenomenon was consistent with previous studies that SDF-1α promoted micro-angiogenesis and facilitated stem cells migration from the microvascular[46–48]. The correlation between MSCs proportion in NP and microvascular in EP made us believe that stem cells in NP of PGD + SDF-1α group mostly migrated from EP and its microvascular. It also represented that the SDF-1α recruited MSCs in NP could only be achieved under the chemokine itself exist and mechanical support from the scaffold simultaneously, and the lack of mechanical support would lead to inward bulging AF and the contents in NP cavity extrusion (Supplementary Fig. 3 NP Injury, HA and HA + SDF-1α group)[22,49]. The MSCs proportion in NP of PGD + SDF-1α group at 16 weeks was 28.46%, which was as much as that of the intact disc (29.32%), PGD group (23.76%), HA + SDF-1α group (27.12%), and higher than that of the NP Injury group and HA group. This phenomenon represented that MSCs recruitment efficacy in NP of PGD + SDF-1α group was decreased at 16 weeks[44]. The disc degeneration level of PGD + SDF-1α group was improved and came to an equilibrium state from the aspect of histological evaluation, disc height, T2 MRI signal intensity of NP, disc mechanical property (Figs. 6 and 7). In contrast, the disc degeneration level of NP Injury group, PGD group, HA group, and HA + SDF-1α group got in worse as time goes by, because of no effective MSCs recruitment in NP during the 16 weeks implantation.

Therefore, the disc height of the PGD + SDF-1α group was maintained and provided enough mechanical support after 1 ~ 8 weeks of implantation, avoiding the extrusion of chemokines SDF-1α then affected the disc regeneration. The NP was regenerated for the recruited MSCs induced by the chemokines[9]. $E_{ins}$ and $E_{eff}$ of the disc in the PGD + SDF-1α group were significantly higher than in the four other groups, and there was no difference with the native intact disc (Fig. 6k, l); the reason was that the PGD NP scaffold with chemokine SDF-1α inhibited the degeneration process of the disc and restored native function of NP. The innovative PGD NP scaffold with regeneration is fabricated based on the optimized thermosensitive shape memory polymer and immobilized with the chemokine SDF-1α. Its biomimetic deformable structure of PGD scaffold achieves the function of minimally invasive implantation. Chemokine SDF-1α immobilizes on PGD NP scaffold induced autologous MSCs to restore NP function and reverse NP degeneration, which will be potential in clinic in the future. There are still limitations in our study. A direct method to demonstrate NP regenerate ability of recruited MSCs is a key point in the following

study, and we will focus on the corresponding in vitro or ex vivo experiments and then processing gene expression analysis of the recruited MSCs in NP in the following study; a radiographic or fluorescent tracer labeled PGD NP scaffold is also desired in the following study to observe its location and degradation after implantation; New Zealand rabbits and Sprague Dawley rats used in our study are very different from the human spine in many aspects. These animals are much more regenerative as the disc contain a high number of notochordal cells and the NP is much more jelly-like than in the human's situation. Further experiments in large experimental animals or human are needed to demonstrate the effectiveness of the PGD NP scaffold described herein.

## Methods

### Preparing PGD with varied synthetic parameters
Glycerin and dodecanedioic acid (Sinopharm) with different molar ratios (1:2, 2:3, 1:1, 4:3) were mixed at 120 °C under nitrogen flow for 24 h. The synthetic product was further reacted under a −0.08 MPa vacuum environment at 120 °C for another 24 h to obtain prepolymer of PGD with different hydroxyl/carboxyl ratios ($MR_{H/C}$: 0.75, 1.00, 1.50, 2.00). The prepolymer was melted and poured into Teflon dish molds to obtain the prepolymer sheet. The prepolymer sheet was further cured in a drying oven under vacuum conditions at −0.1 MPa and 120 °C, then collected as an experiment sample every 24 h until 216 h. Carefully remove the cross-linked PGD from the mold, and a laser cutting machine (VLS2.30, Universal) was employed to cut the polymer as the required shape.

### Thermodynamic properties of PGD
Differential scanning calorimetry (DSC8000, Platinum Elmer) was used to test the thermodynamic properties of the cured PGD sample with different synthetic parameters ($MR_{H/C}$ and cure time). About 2 mg of sample was weighed and placed in an aluminum crucible. The crucibles were preheated to 80 °C and held for 1 min to remove heat history. A temperature scanning program was employed from 80 °C to −30 °C to 80 °C at the rate of 10 °C/min. The heat flow of the sample was recorded by Pyris software (version 5.0, Platinum Elmer), and the results were calibrated using peak and enthalpy tools.

### Shape memory properties of PGD
A bending-unfolding method was used to measure the shape fixation and recovery abilities[50]. The rectangular samples ($16 \times 5 \times 1$ mm) were immersed in a 40 °C water bath for 5 min, fixed in a 3D printed U-shape mold, then placed at room temperature for about 10 min to wait for cooling and shaping. A digital image analysis device (CARE) was used to measure the angle at both sides of the specimen, recorded as $\theta_{max}$. Take the sample from the mold and measure the angle, recorded as $\theta_{fix}$. Immerse the sample in a 37 °C-water bath, record its unfolding process and measure the final angle as $\theta_i$. Calculate the shape fixation rate ($R_f$) and shape recovery rate ($R_r$) of the specimen by the following formulas: $R_f = \theta_{fix}/\theta_{max}$, $R_r = (\theta_{max} − \theta_i)/\theta_{max}$. Experimental data analysis and plotting were applied by Excel software (version 2019, Microsoft). SPSS software (version22, IBM) was used for statistical analysis.

### Mechanical properties of PGD
A universal testing machine (IPBF series, CARE) equipped with a 100 N load cell and temperature-controlled water bath was used to measure the mechanical property of the material/scaffold. (1) Young's modulus of PGD with varied synthetic parameters was obtained following the American society for testing and materials (ASTM) -D638 standard. PGD sheets with 1 mm thickness were laser cut into $35 \times 6 \times 1$ mm dog-bone-shaped specimens. The specimens ($n = 6$) were stretched to fracture at a 37 °C water bath at the speed of 1 mm/min; (2) Young's modulus of the vortex-plate-shaped scaffolds fabricated by PGD with three different synthetic parameters (1.50 $MR_{H/C}$/72 h, 1.50 $MR_{H/C}$/

120 h, 1.50 $MR_{H/C}$/168 h) was measured. The vortex-plate-shaped scaffolds with about 4 mm diameter and 1 mm thickness were obtained by cutting PGD sheets. The scaffolds were compressed to 0.1 mm at 37 °C water bath with 0.01 mm/min speed. Young's modulus of PGD NP scaffold was calculated according to the standard of ASTM-D6108; (3) Four steps compressive stress relaxation experiment was performed to evaluate the mechanical function of the lumbar disc. Vertebral-disc samples were collected from ex vivo or in vivo experiments. Peripheral muscle, spinal cord, spinous, and transverse processes of the vertebral-disc samples were removed. Residual vertebral bodies between the disc were embedded with commercial modified acrylate with fast cure and high strength abilities (GLH corporation, Fushun China). A stress relaxation program was set up for the treated samples to be compressed to 5% strain, followed by 360 s of relaxation. The process was repeated until 20% strain of the initial disc height. The stress ($\sigma$)-time ($t$) data was fit to a poroelastic model using a MATLAB script as the following formula described: $\sigma = A (1 − e^{-t/\tau}) + B$, where $\tau$ represents the time constant of relaxation. The disc's effective equilibrium modulus ($E_{equ} = A + B$) and instantaneous modulus ($E_{ins} = B$) were then calculated according to the linear fitting results[51].

### Cell proliferation of PGD
Balb/C 3T3 cells were cultured on the polymer substrates to evaluate biocompatibility. Briefly, $5 \times 10^4$ cells were seeded on the polymer sheets at a 24-well plate and cultured in an incubator (37 °C, 5% $CO_2$). DMEM medium was replaced every other day. Samples were collected 1, 3, and 7 days after culture, and a CCK8 kit (Biorigin) tested the cell proliferation ability. Then, the surface morphology of the collected samples was observed through SEM (Quanta FEC250, FEI) after 2.5% glutaraldehyde fixation and ethanol dehydration treatment.

### In vivo degradation of PGD in rat subcutaneous implantation
In vivo experimental procedures were performed following the Beihang University's Committee on Use and Care of Animals (Permit No.BM201900084). Male Sprague-Dawley rats ($n = 6$) weighing 250–300 g were anesthetized with sodium pentobarbital. PGD Samples (1.50 $MR_{H/C}$/72 h) were cut into rectangular pieces ($16 \times 5 \times 1$ mm) by laser and sterilized under 121 °C steam for 20 min. A longitudinal incision was made at the rat's back midline, and the pocket was bluntly separated. Samples were carefully placed in the pocket, and the incision was closed with surgical sutures. The rats were sacrificed through $CO_2$ asphyxiation and cervical dislocation at 1 week, 4 weeks, and 9 weeks after implantation. Then, the implanted samples ($n = 6$) with peripheral tissue were collected and fixed with 4% paraformaldehyde solution. The collected PGD samples were removed from the peripheral tissues, cleaned, dried, and weighed to evaluate the degradation properties of the polymer.

### Numerical simulation of the PGD NP scaffold
Quasi-static model was developed to measure the unconfined-compressive modulus of the scaffold. The vortex-plate-shaped scaffold model with about 4 mm diameter and 1 mm thickness designed by computer-aided design software was converted to an ASC file and imported into the ABAQUS (version 6.14, Dassault Systèmes) project. Over 15000 hexahedral elements were created on the scaffold model by meshing tools. Mechanical properties of PGD with twelve synthetic parameters with four $MR_{H/C}$ and three cure times, including elastic modulus, plastic deformation, and Poisson's ratio were assigned to the scaffold model. Young's modulus of the scaffold was calculated based on the numerical simulation output. In addition, the finite element model of the vertebra-disc was developed based on Micro-CT (Sky-Scan1276, Bruker) images of the New Zealand rabbit lumbar segment with 20 μm pixel resolution and 100 μm slice thickness. CT image segmentation, geometric model reconstruction, mesh division, material property assignment, and boundary/loading conditions settings

were performed respectively to establish the 3D numerical simulation model including vertebrae body, EP, NP, and AF as shown in Fig.3a, b. The force of 10 N was applied at the upper surface of L5 endplate while L6 vertebra was fixed completely. The native intact disc model, and disc models with implanted PGD and HA NP scaffolds separately were established as in Fig. 3c–e. A hole with a diameter of 1.2 mm was designed in the right center of the disc to simulate the punctured hole after implanted. Material parameters of the vertebral body, intervertebral disc, PGD, and hydrogel in the model were listed in Supplementary Table 5[52–54].

## Fabricating PGD NP scaffolds for NP implantation

Vortex-plate-shaped PGD NP scaffolds with about 4 mm diameter and 1 mm thickness were designed according to the rabbit's NP and manufactured by the laser cutting machine[55]. The scaffolds were cleaned in normal saline. Shape programming was performed by stretching the vortex-plate-shaped scaffold into a rod shape (Supplementary Fig. 6 and Supplementary Movie 5). Platinum wire weighing 1 mg was fixed around the head of the rod-shaped PGD NP scaffold as a developing ring for intraoperative fluoroscopy (Fig. 1c). The NP scaffold was inserted into a stainless-steel syringe needle with 1.2 mm diameter and sterilized at 121 °C for 20 min before implantation.

## Immobilizing SDF-1α to PGD and HA NP scaffolds

Human recombinant SDF-1α (Novoprotein) was immobilized onto the PGD NP scaffolds through heparin-mediated interaction[43]. In detail, $NH_2$-PEG-$NH_2$ was used as a linker molecule to covalently connect the scaffold and heparin via EDC and NHS (Sigma-Aldrich). According to specific heparin-mediated interaction, the modified PGD NP scaffolds could absorb SDF-1α by soaking in a phosphate buffered saline (PBS) solution of 1000 ng SDF-1α at 4 °C for 12 h. Hyaluronic acid (Biorigin) scaffolds with SDF-1α were taken as the control group for the in vivo experiment. HA powder was sterilized under an 18 W UV lamp for 2 h and added to 1 ml PBS solution dissolved with SDF-1α (20 μg/ml). HA gel with SDF-1α was obtained after stirring the mixture thoroughly. The gel was then centrifuged to remove bubbles, and 50 μl of it was packed into micro-syringes for surgery. The loading efficiency and in vitro release ability of immobilized SDF-1α were evaluated under the following 7 days of static conditions. Scaffolds were soaked in 1 ml PBS at 37 °C. 200 μl PBS was removed and supplemented with equal volumes of the liquid at each time point. The cumulative release amount of SDF-1α in PBS was measured by an ELISA kit (Neobioscience) and SkanIt colorimetric software (version 7.0, ThermoFisher).

## Surgical preparation of animal experiments

In vivo experimental procedures were performed following the Beihang University's Committee on Use and Care of Animals (Permit No.BM20200146). New Zealand white rabbits weighing 2.5–3.5 kg at 6–8 months of age were selected as experimental animals for L5-L6 lumbar disc puncture following NP scaffold implantation. The NP scaffolds were divided into four experimental groups, including PGD, PGD + SDF-1α, HA, and HA + SDF-1α (Fig. 1c). The NP Injury disc from the rabbits with only puncture treatment was proceeded as pre-experiment to validated NP injury efficiency when most of the AF remained intact, namely NP injury group. The native intact disc from the rabbits without surgical treatment was used as the negative control, namely the native intact group. Eight rabbits of each group were used as parallel animal samples to implant the NP scaffold. Four were sacrificed 8 weeks after implantation to harvest their L5-L6 discs, and another four were treated at 16 weeks.

## The in vivo L5-L6 implantation process

A stainless-steel needle with 1.2 mm diameter was punctured into L5-L6 nucleus cavity of the rabbit. The puncture position of the needle was fine-tuned with the help of an intraoperative C-arm machine. The NP

was sucked out through a syringe. For NP Injury group, incision suture was carried out with no further implantation operation. For PGD and PGD + SDF-1α group, a rod-shaped NP scaffold was inserted from the needle's tail and gently pushed into the nucleus cavity. Body temperature triggered shape transform of PGD NP scaffold was accomplished. We can observe if deliver process was successful by the developing ring of PGD NP scaffold under intraoperative X-ray fluoroscopy. For HA and HA + SDF-1α group, 50 μl of the hydrogel was slowly injected into the nucleus cavity. The needle was then pulled out, and the surgical incision was sutured carefully. Each postoperative rabbit was cared for in a single cage and followed closely to avoid abnormal pain or discomfort.

## T2 MRI signal intensity and DHI of the surgical disc with implanted four scaffolds

The rabbits were anesthetized and performed magnetic resonance imaging after 1, 4, 8, 12, and 16 weeks of implantation. T2 weighted MRI images along the coronal axis of the lumbar were obtained by a 3 Tesla MRI scanner (TRIO 3 T, Siemens). The T2 signal intensity of L5-L6 disc was measured using a DicomGo viewer (version 2.0.1.4, Link Imaging) and normalized with that of the native intact disc from the same spine[56]. The disc height was measured by sagittal X-ray fluoroscopy images obtained by a C-arm machine (OEC Touch, GE) after 1, 4, 8, 12, and 16 weeks of implantation. DHI of L5-L6 disc was calculated and normalized to native intact disc from the same spine as described in Supplementary Fig. 7[57].

## Histological analysis of the disc with implanted four scaffold groups

The collected vertebral-disc samples were fixed in neutral formalin solution, decalcified in 0.15 M EDTA solution, embedded in paraffin, and then sectioned into 4 μm thickness tissue slices along the coronal plane. Histological staining was used to analyze the slices of each sample by HE, Mason's trichrome, Alcian Blue & nuclear fast red, Safranin O & fast green staining, respectively (Fig. 6a and Supplementary Fig. 8). Histological staining images were taken by an optical microscope with CCD camera (IX70, Olympus) and cellSens software (version 1.7, Olympus). Images adjustment and analysis were operated by ImageJ software (version 1.6.0, NIH). Histological grading of disc degeneration was compared with each group's disc according to evaluation scale as shown in Supplementary Table 2[37]. In detail, Samples were scored from five aspects: cellularity of AF, the morphology of AF, AF-NP border, cellularity of NP, and morphology of NP. The histological scores for each group were evaluated via HE, Masson, Alcian Blue, and Safranin O staining images from four rabbits' samples. The number of microvascular near the border between EP and NP was obtained by manually counting from digitally stitched HE images with the whole morphology of the disc.

## Immunofluorescence analysis of disc with implanted four scaffold groups

Immunofluorescence staining was used to analyze the distribution of MSCs in the intervertebral disc. Antigen retrieval was performed by soaking deparaffinized sections in a solution containing sodium citrate for 20 min at a 90 °C water bath. The slices were blocked with 5% BSA solution, incubated with 1:10 diluted CD90 and CD166 primary antibodies (ab225 and ab235957, Abcam) for 12 h at 4 °C, followed by incubation with 1:200 diluted Alexa-fluor 488 and Alexa-fluor 647-labeled secondary antibodies (A32723 and A32733, ThermoFisher) for 1 h at room temperature. DAPI staining was used to display the cell nucleus. Immunofluorescence staining images were observed by confocal microscope (SP5, Leica), and analyzed by LAS X software (version 3.7.4.23463, Leica). For each immune stained section, the cells in the bone marrow and growth plate of vertebra were used as positive controls for CD90 & CD166 co-positive cells, while osteocytes in EP

were used as negative controls. The number of CD90 & CD166 co-positive cells in NP cavity of each sample was counted based on the 20× Immunofluorescence images from three parts of the nucleus cavity. At least four parallel animal samples were used for each data of the experimental groups.

## Reporting summary

Further information on research design is available in the Nature Portfolio Reporting Summary linked to this article.

## Data availability

The experimental data and their statistical analysis generated in this study are provided in the Supplementary Data 1-6, including histological analysis, microvascular in EP, MSCs proportion in NP, DHI variation during 16 weeks implantation, MRI T2 signal intensity variation during 16 weeks implantation, and disc mechanical properties after implantation.

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

## Acknowledgements

Y.F. and L.W. were supported by the National Natural Science Foundation of China (T2288101, 11822201, 11827803, 12172034, U20A20390). L.W. and K.J. acknowledged Beijing Municipal Natural Science Foundation (7212205) with financial support for this paper. Y.F. was supported by the 111 project (B13003). L.W. was supported by the Fundamental Research Funds for the Central Universities.

## Author contributions

L.W. and K.J. contributed equally to the work. Designed the research: L.W., Y.F., and S.J.H. Materials PGD synthesized: L.W., K.J., H.R., and S.J.H. Mechanical test and data analysis: K.J., P.X., L.W., and Y.F. Biodegradation experiments and data analysis: K.J, L.W., Y.F. Biocompatible experiments and data analysis: K.J., L.W., H.Y., and Y.F. In-vivo implantation experiments and care for rats and rabbits: K.J., N.L., H.Y., and L.W. Histological observations and data analysis: K.J., H.Y., N.L., and L.W. Finite element model develop and data analysis: L.W., P.X., K.J., and Y.F. MRI images observations and T2 signal intensity analysis: N.L., K.J., and Y.F. Manuscript writing and editing: L.W., K.J., N.L., P.X., H.Y., H.R., S.J.H., and Y.F.

## Competing interests

The authors declare no competing interests.

## Ethics

We have complied with all relevant ethical regulations declared in the manuscript, and disclosed the name(s) of the board and institution in the methods part.
