## [Peer Review File · Nature Communications]

Innovative design of minimal invasive biodegradable poly (glycerol-dodecanoate) nucleus pulposus scaffold with function regenerationREVIEWER COMMENTS

Reviewer #1 (Remarks to the Author):

Authors have prepared a invasive biodegradable scaffold to repair and regenerate NP with functionality. The design of the scaffold is inspired by cucumber tendril using shape memory polymer poly(glycerol-dodecanoate). Author were mainly focused on mimicking the mechanical property with that of human nucleus pulposus by adjusting synthetic parameters. Work is of great relevance in the field of IVD regeneration and its management. However, lack of appropriate control and implementation problem are not well discussed. Some major concerns are raised in below comments related to the data presented in this work. Major revision is required to support the aim of this manuscript.

1. Injury group is missing in fig. 5, 6 and 7, How the injury was created when the outer AF remain intact as shown in figure 6. Also, inserting .2 mm injury is big to appear on H&E data and model should be validated before implantation.
2. Figure 7 showed healing of NP region at 8 weeks without any sign of PGD scaffold but according to degradation data approximately 50-60% of scaffold remained after 35 days. Author may like to confirm the presence of scaffold at implantation in concurrence with the NP regeneration and cellular repopulation. H&E study does not show any scaffold at the site of injury. Is the scaffold degraded by 8 week? It is important to show the scaffold after the implantation may be at day 0 or day 7 or 14. It is also important to confirm the shape and orientation of injected scaffold as claimed by authors.
3. In fig. 5, no significant difference was observed in the expression of CD90 and CD166, hence the recruitment of stem cells may not be dependent on sdf1 α , or may sdf1 α is non-functional or is not released to have sufficient chemotaxis.
4. Also, the characterization of stem cell differentiation in to NP cells is desirable.
5. In figure 6, it looks like the organization of AF is completely lost after 16w of implantation in PGD group compared to 8 week. Such AF organization is also not clear in intact IVD and other treatment group. It showed that staining is not uniform.
6. In figure 7, the disc height of intact IVD before implantation looks different, also the way it is represented demonstrate only the after the implantation of scaffolds. Also, consistency in DHI after making defect is required to evaluate the effect of PGD scaffold.
7. It is demonstrated that curling of scaffold occurs in water in vitro at 37 degree (supplementary video). However, in the in vivo study the NP was sucked out and hence there is not enough water at the site. The should explain if the PGD rod can curl in the absence or minimal presence of moisture. Curling of scaffold in air at 37C should be demonstrated.
8. Pain markers should also be tested post-implantation to confirm the functional repair.
9. Magnified and clear images of AF and NP cellularity and morphology should be included in the manuscript to correlate the presented data.
10. The developed material is non-porous in nature and the sdf1 α will be presently mostly on the surface of the device. It means most of the stem cells will be recruited on the surface of the implant material. Then how they reach to the bulk of the material after implantation as shown in H&E and other images. Also, what type of stem cells will be recruited and how? IVD is mostly avascular tissue and thus

may not be able to recruit cells from blood. Does it recruit cells from the blood supply in the endplates and how? It should be discussed with reference in the discussion.

Reviewer #2 (Remarks to the Author):

In this paper, the shape memory polymer poly(glycerol-dodecanoate) scaffold with chemokine stromal cell-derived factor-1 α was applied in treating intervertebral disc degeneration. And the major conclusion is that this treatment improves the disc height and recruit autologous stem cells.

The data is not sufficient for the conclusion. In addition, the novelty of PGD scaffold with SDF-1 is limited.

Here are some questions:

1. Why the PGD scaffold with SDF-1 has significant regenerative effects on degenerated NP, just by providing mechanical property and recruiting autologous stem cells? How many MSCs were recruited to the NP and survived after 16 weeks? More evidences should be provided to make readers convinced.
2. Why CD90+&CD166+ cells were decreased after 16W compared with 8W in the PGD+SDF-1 group, but the regenerative effects were increased?
3. In Fig.6, the NP was obvious in the intact group, but the area of NP could not be distinguished with AF after 16W in each group. There is a significant difference between healthy NP and PGD+SDF-1 treated NP.

Reviewer #3 (Remarks to the Author):

The authors present a tissue engineering study for the application of a biomaterial that can keep its shape in a “memory”. Furthermore, the material allows releasing a growth factor, in this case, SDF-1 α . This growth factor is known to recruit mesenchymal stromal cells (and other progenitor cells) and is important for the field of regenerative medicine and the field of orthopaedic research on the spine. The manuscript is on the innovation of shape-memory materials that could find their way to clinical application.

The study is supported by in-vitro biomechanical characterization of the material and by an in-vivo rabbit model for up to 9 weeks subcutaneously. Biocompatibility of the material was also tested in a subcutaneous model in Sprague-Dawley rats.

Finally, the animal model (New Zealand white rabbit) is back-upped by numerical simulations of the PGD NP scaffold.

Abstract: The abbreviations should be already introduced in the abstract, i.e., SDF-1 α and PGD as the

abstract should be independent of the main text.

Page 6 line 109: please, recall for the reader the young's modulus (compressive Modulus) for a healthy non-degenerated IVD (for both, NP and AF). Possibly, also provide a reference for it.

Line 374: Please, specify the standard of ASTM. For what does ASTM stand for?

Line 378: Please, provide brand and city of manufacturer of modified acrylate, also specify in what way the acrylate was modified.

Line 416: please, add scanning resolution of the μ CT and slice thickness of the scans.

Line 492: Four parallel rabbit samples graded the histological scores for each group via HE, Masson, Alcian Blue, and Safranin O staining images. \diamond strange wording, rephrase this sentence

Line 501: Please, provide working concentrations of CD90 and CD166 primary antibodies and secondary antibodies. Lines 504 and the following: describe your negative and positive controls for the immune staining.

Figure 6 b and c: with N = 4 data points per group the bar plots should be replaced with single data point plots and overlaying with means \pm SD or SEM.

Figure 7 k and l: also, here replace bar plots with single data plots and overlay means \pm SEM.

Discussion

The authors should include a discussion on the limitations of the rat and the rabbit model for IVD research. These animal groups are very different from the human spine in many aspects. Most importantly, these animals are much more regenerative as the IVDs contain a high number of notochordal cells and the NP is much more jelly-like than in the human situation.

The figures and the legends in general are fine. However, following the trend of showing all the data points in the graphs I recommend the authors to switch from bar plots to single data graphs.

The authors do show very nice histology using paraffin. For paraffin, the tissue needed to be decalcified prior to embedding. Are the authors not afraid that this step would generate artefacts as also proteoglycans are washed out during this step? Would not hard sections be the better option?

Fig.6 should add in the caption that these are paraffin sections.

Point-to-point responses to the Reviewers' comments

Reviewer #1 (Remarks to the Author):

Authors have prepared a invasive biodegradable scaffold to repair and regenerate NP with functionality. The design of the scaffold is inspired inspired by cucumber tendril using shape memory polymer poly(glycerol-dodecanoate). Author were mainly focused on mimicking the mechanical property with that of human nucleus pulposus by adjusting synthetic parameters. Work is of great relevance in the field of IVD regeneration and its management. However, lack of appropriate control and implementation problem are not well discussed. Some major concerns are raised in below comments related to the data presented in this work. Major revision is required to support the aim of this manuscript.

1. Injury group is missing in fig. 5, 6 and 7, How the injury was created when the outer AF remain intact as shown in figure 6. Also, inserting .2 mm injury is big to appear on H&E data and model should be validated before implantation.

Response:

Thanks for the suggestion. We have added images/data of the NP Injury group with injured NP and intact outer AF in Fig. 5, Fig. 6, and Fig. 7, which facilitates to observe disc degeneration at cellular composition, histomorphology, and clinical indexes, and to compare the differences with the intact group and other NP scaffold groups. The operational process of the NP Injury group is as same as other NP scaffold groups except for inserting an NP scaffold. As shown in the enlarged AF-NP boundary images in Fig. 6a, we have added the description in detail in revised manuscript at Page 9 line 178: "For the NP injury group, AF maintained its concentric circles morphology as intact disc but with serpentine fibers in it at 8 weeks, AF appeared inward bulging and the disc height decreased at 16 weeks, fibrosis in NP cavity and osteophyte in outer AF were observed at the same time". From the aspects of clinical indexes (Fig. 7) and cellular composition (Fig. 5), Disc height and T2 MRI signal intensity of the NP Injury group is significantly reduced at 8 weeks, the number of NP cells is lower than the intact one and most

of their nucleus showed flattened in a typical apoptotic state. Disc degeneration gets in worse over time, Disc height and NP MRI signal intensity of NP Injury group also decrease at the same time, as shown in Fig. 7h, i.

Fig. 5 Stem cells recruitment after 8, 16 weeks implantation of PGD, PGD+ SDF-1 α , HA, HA+SDF-1 α scaffolds in L5-L6 disc. **a**, Histological images including HE and immunofluorescence staining images of EP in the coronal plane after 8 weeks implantation for native intact disc, NP Injury disc, PGD, PGD+SDF-1 α , HA, HA+SDF-1 α scaffolds. White dotted line represented the boundary between EP and NP, the black/white arrows indicated micro-vessels in EP, and the green dotted circle represented migrated chondroid cells from EP. **b**, Histological images including HE and immunofluorescence staining images of NP cells in the coronal plane after 8 weeks implantation. **c, d**, Number of micro-vessels in EP after 8 weeks (**c**)(n=4) and 16 weeks (**d**)(n=4) implantation for native intact disc, NP injury disc, and above mentioned four scaffold groups. **e, f**, Percentage of

CD90 and CD166 positive cells in NP after 8 weeks (e)(n=4) and 16 weeks (f)(n=4) implantation for native intact disc, NP injury disc, and above mentioned four groups. Transverse lines indicated statistical differences between two groups analyzed by independent-samples t-test ($P < 0.05$).

Fig. 6 Histological paraffin sections images and evaluation of disc degeneration after 8, 16 weeks implantation. **a**, HE, Masson and Alcian blue stained images of native intact disc, NP injury disc, and disc implanted with above mentioned four scaffolds in the coronal plane at 8 and 16 weeks. Blue dotted circle represented PGD scaffold in NP. Black dotted box indicated areas of osteophyte

generated. Red dotted lines represented AF outline of the disc. **b, c**, Histological scores of disc degeneration for native intact disc and four experimental groups at 8 weeks (**b**)(n=4) and 16 weeks (**c**)(n=4). Transverse lines indicated statistical differences between the two groups determined by the independent-samples t-test ($P < 0.05$).

Fig. 7 Disc height, T2 MRI signal intensity and mechanical recovery of discs with PGD, PGD+SDF-1 α , HA, HA+SDF-1 α scaffolds. **a**, X-ray and MRI T2 images L5-L6 discs for native Intact

disc, NP Injury disc, and four experimental group before and after 8,16 weeks implantation (anatomical directions: H-Head, F-Foot, V-Ventro, D-Dorsal). The yellow arrow represented platinum ring on PGD scaffold. **b, d, f**, Normalized DHI of L5-L6 disc in NP injury disc and four experimental groups after 1 week (**b**)(n=4), 8 weeks (**d**)(n=4) and 16 weeks (**f**)(n=4) implantation. **c, e, g**, Normalized T2 signal intensity of L5-L6 disc in nucleus cavity of NP injury disc and four experimental groups after 1 week (**c**)(n=4), 8 weeks (**e**)(n=4) and 16 weeks (**g**)(n=4) implantation. **h**, Relationship of normalized DHI and implanted time for NP injury disc and four experimental groups. **i**, Relationship of normalized T2 signal intensity of L5-L6 disc in nucleus cavity with implant time for NP injury disc and four experimental groups. **j**, Loading test program of four-step compression stress relaxation for the collected L5-L6 discs after 16 weeks implantation. The collected discs were removed the peripheral muscle, spinal cord, spinous process, and transverse process as shown in the upper left picture. **k, l**, Effective instantaneous (**k**), and equilibrium moduli (**l**) were calculated based on stress-strain curves (n=4). Transverse lines indicate statistical differences between the two groups determined by the independent-samples t-test ($P < 0.05$).

Φ1.2 mm injury is almost as big as L5-L6 disc height observed from HE images at coronal plane, and it is the optimized puncture diameter evaluated by our previous *in vivo* pre-experiments after weighing the surgical operability against physical dimension of PGD scaffold. A smaller puncture diameter less than 1.2 mm makes it difficult to suck jelly-like NP during surgery and also limits the height of the implanted PGD scaffold to provide effective mechanical support after implantation. A larger puncture diameter over 1.2 mm makes the syringe needle difficult to insert into the space between vertebrae accurately. The exposure area of rabbit L5-L6 discs along the sagittal plane of the spine is far smaller than that of humans, which makes it more difficult to locate and operate than clinical surgery. To insert the 1.2 mm needle smoothly, we use a self-made operating table with 60° angle and fix the rabbit with lateral position, thus allowing a greater exposure area of the L5-L6 disc by bending the rabbit's lumbar (Fig. R1-1). Besides, it is also good for closure the punctured hole in AF after unloading the bended lumbar, as evidenced by the complete AF morphology with concentric circles for the NP Injury group at 8 weeks, as shown in Figure 6a. We have proceeded with *ex vivo* and *in vivo* pre-experiments before NP scaffold implantation, including needle size optimization, surgical approach exploration, intraoperative navigation, and animal model verification. The added experimental data of the NP Injury group are obtained from the results of the pre-experiments.

Figure R1-1. Surgical process before inserting the puncture needle.

2. Figure 7 showed healing of NP region at 8 weeks without any sign of PGD scaffold but according to degradation data approximately 50-60% of scaffold remained after 35 days. Author may like to confirm the presence of scaffold at implantation in concurrence with the NP regeneration and cellular repopulation. H&E study does not show any scaffold at the site of injury. Is the scaffold degraded by 8 week? It is important to show the scaffold after the implantation may be at day 0 or day 7 or 14. It is also important to confirm the shape and orientation of injected scaffold as claimed by authors.

Response:

Thanks for the comments. To confirm the delivered PGD scaffold into NP cavity, several coils of platinum wire were wrapped around one end of the rod-shaped PGD scaffold prior to surgery to allow for real-time observation of its placement during the operation by X-ray machine, as shown in Fig. 1c. We also confirm the position of PGD scaffolds at 1 week, 4 weeks, 8 weeks, 12 weeks, and 16 weeks after implantation through the platinum marker in X-ray images, as the yellow arrows shown in Fig. 7a. PGD scaffold and surrounding disc in the paraffin embedding blocks can also be clearly distinguished as shown in Fig. R1-2, which is reflected as the transparent areas in the histological stained sections, as the blue dashed box shown in Fig. 6a. With the PGD scaffold degradation completely in NP, it no longer exists in the histological stained sections at 16 weeks. This phenomenon is consistent with the results of our subcutaneous degradation experiments (Fig. 4c), and PGD loses about 80% of its initial mass at 8 weeks after implantation and degrades completely at 9-10 weeks. Gradual degradation of PGD scaffold in NP ensures mechanical support provided from the scaffold at the first few

weeks and releases space occupy of the scaffold when its degraded, which is beneficial to the reconstruction of tissue engineering.

Figure R1-2. PGD scaffold and surrounding disc can be clearly distinguished in the 8 weeks paraffin embedding blocks, which is reflected as the transparent areas in the histological stained sections.

3. In fig. 5, no significant difference was observed in the expression of CD90 and CD166, hence the recruitment of stem cells may not be dependent on $sdf1\alpha$, or may $sdf1\alpha$ is non-functional or is not released to have sufficient chemotaxis.

Response:

Thanks for the comments. Fig. 5 is modified to present tissue and cell morphology in EP-NP boundary (Fig. 5a) and inner NP (Fig. 5b) respectively. NP Cells are presented with high magnification HE images in Fig. 5b, and cellularity among Intact group, NP Injury group, and four NP scaffold groups shows differences from each other. According to previous study, normal NP cells shows stellar shape surrounded with dense eosin-stained extracellular matrix; apoptotic NP cells always shows flat shape; notochord-derived cells or bone marrow-derived stem cells always shows rounded shape with rich cytoplasm.^{1,2,3,4} It is confirmed in our study by HE-stained images and immunofluorescence images, as shown in Fig. 5b. We find that stem cells from the bone marrow and growth plate of the vertebral tend to be CD90 & CD166 co-positive cells with round shape and rich cytoplasm, while NP cells with stellar and flat shape in the intact disc cannot be stained by CD90 or CD166 simultaneously.

It should be noted that microvascular in EP of PGD+SDF-1 α group is significantly higher than that of PGD group without SDF-1 α and other groups at 8 weeks (Fig. 5c), which is consistent with previous mentioned SDF-1 α promoting vascularization.^{5,6,7} The microvascular in EP of

HA+SDF-1 α group is not higher than that of HA group, this phenomenon is related to SDF-1 α extrusion after implantation. The extrusion also promotes the aggregation of chondroid cells at outer AF and promotes the osteophytes formation in HA+SDF-1 α group (Fig. 6a and Supplementary Table 1), which is consistent with the results of previous studies.⁸

In general, SDF-1 α in our study plays a key role in regulating distribution of stem cells in NP and microvascular in EP, as well as the formation of osteophytes at outer AF. Releasing SDF-1 α in NP without extrusion will promote regeneration process of the injured NP.

Groups	8 Week	16 Week	Total osteophyte incidence
Intact	0.0%	0.0%	0.0%
NP Injury	50.0% (2/4)	100.0% (4/4)	75.0% (6/8)
PGD	0.0% (0/4)	25.0% (1/4)	12.5% (1/8)
PGD+SDF-1 α	0.0% (0/4)	25.0% (1/4)	12.5% (1/8)
HA	25.0% (1/4)	75.0% (3/4)	75.0% (6/8)
HA+SDF-1 α	75.0% (3/4)	75.0% (3/4)	75.0% (6/8)

Supplementary Table 1. Osteophyte incidence in the disc of Intact, NP Injury, PGD, PGD+SDF-1 α , HA, HA+ SDF-1 α group.

4. Also, the characterization of stem cell differentiation in to NP cells is desirable.

Response:

Thanks for the comments. Cell therapy is widely researched in degenerated discs treatment in clinic, among which MSCs are the most commonly used cell type.⁹ As the description described in revised manuscript at Page 14 line 302:“ The mechanism of MSC promoting disc repair was still unclear. Current studies believed that tissue repair function of MSC was closely related to its secretion and extracellular vesicles mediated inhibitory effect of inflammation and apoptosis, and cell differentiation effect.^{10, 11, 12} Mechanical environments had also been shown to affect differentiation effect of MSC into NP cells through the YAP1 signaling pathway”.¹³ Besides, Microenvironment of the degenerated tissue can also promote the inside cells secreting

chemokines to recruit MSC in the surrounding tissues and peripheral vascular to realize tissue regeneration.^{14, 15} Avascular tissues like NP cannot repair itself after degeneration due to the limited stem cells recruit ability.¹⁶ Recent studies have shown that delivering SDF-1 α in NP significantly promotes the migration of autologous MSCs and restores degenerated disc function in the non-stress load disc model.¹⁷

In our study, an SDF-1 α delivery PGD scaffold with minimally invasive and mechanical support ability is developed based on its stem cell recruiting ability above mentioned, which achieves NP regeneration with untreated AF wound and daily disc load after implantation. PGD scaffold realizes the expect well from the results of postoperative tissue morphology, disc height, NP MRI signal and mechanical property, compared with the NP injured group and HA scaffolds (Fig. 6 and Fig. 7). It indicates that the recruitment of stem cells has achieved the positive repair effect on the injured NP.

5. In figure 6, it looks like the organization of AF is completely lost after 16w of implantation in PGD group compared to 8 week. Such AF organization is also not clear in intact IVD and other treatment group. It showed that staining is not uniform.

Response:

Thanks for the suggestion. NP-AF boundary morphology in Fig. 6 is modified with high contrast black and white images to reflect AF organization clearly at 8 weeks and 16 weeks among the Intact, NP Injury, and four NP scaffold groups. Also, the red dashed curves are used to mark AF organization in Fig. 6a to allow a more obvious visibility of morphological changes during 16 weeks implantation. The description is added in revised manuscript at Page 9 line 178 as following: “As shown in Fig. 6a, the intact disc was found to have a clear NP-AF boundary and well-organized collagen lamellae without ruptured or serpentine fibers, and no fibrotic tissue was observed in NP from the Masson-stained images, blue proteoglycan was rich in surrounding cells in NP which were homogeneously distributed as shown in Alcian Blue-stained images; For the NP injury group, AF maintained its concentric circles morphology as

intact disc but with serpentine fibers in it at 8 weeks, AF appeared inward bulging and the disc height decreased at 16 weeks, fibrosis in NP cavity and osteophyte in outer AF were observed at the same time; For the PGD group, a PGD NP scaffold was observed in the nucleus cavity, which means that it did not wholly degrade at 8 weeks, and it was similar to native intact disc in Masson and Alcian Blue staining. PGD NP scaffold was degraded entirely at 16 weeks, and prominent serpentine fibers were observed in AF. Fibrosis occurred in NP, and disc height decreased. No obvious proteoglycan was observed in NP; For the PGD+SDF-1 α group, the PGD NP scaffold was also not completely degraded at 8 weeks, and it showed slight degenerative features compared with the intact group, including serpentine fibers in AF and misty NP-AF boundary. And there was no obvious variation for the NP and AF at 16 weeks; For the HA group, inward bulging of AF was observed clearly from HE-stained images at 8 weeks, which led to disc height decreasing and the nucleus cavity compressed. AF inserted into the nucleus cavity was found, and more than one-third of AF bulged inward, obviously at 16 weeks. Collagen I was distributed in NP from Masson-stained images, which means severe fibrosis of NP, and osteophytes formed surrounding the punctured hole in AF; For the HA+SDF-1 α group, inward bulging of AF was also observed from HE-stained images at 8 weeks. Collagen I was rich in NP based on Masson-stained images, and osteophytes also formed surrounding the punctured hole in AF, and Collagen I in NP increased at 16 weeks, but proteoglycan was hardly found in NP”.

6. In figure 7, the disc height of intact IVD before implantation looks different, also the way it is represented demonstrate only the after the implantation of scaffolds. Also, consistency in DHI after making defect is required to evaluate the effect of PGD scaffold.

Response:

Thanks for the suggestion. We noted the difference in initial disc height among the groups you mentioned in Figure 7a, and average disc height of disc samples used in our study is 1.81 ± 0.17 mm ($DHI = 8.38E-2 \pm 7.82E-3$). It has a little difference among each sample, and the difference should come from individual differences among rabbits even though their weight and age are

screened before surgery. However, normalized DHI (current DHI / initial DHI) was used to eliminate the individual differences among rabbits for presenting the disc height variation after implantation.

Consistent clinical indicators should be confirmed to reflect similar initial NP damage among all surgical treated groups, to compare subsequent disc degeneration level between each group. The description in detail is described at revised manuscript at Page 15 line 311: “T2 MRI signal intensity of NP at 1 week after surgical treatment was used as the clinical indicator to reflect NP damage among each group in this study, and they were all ~60% of their initial state before surgery with no statistical difference”. We believe that DHI after surgical treatment is not as good as T2 MRI signal intensity at 1 week for the indicator, because nearly no changes are happened to DHI without stress load after surgery. DHI is better to reflect mechanical support ability of the disc after surgical treatment among each group. As shown in Fig. 7h, DHI will only be maintained under stress load at 1 week after surgery when its NP function well or PGD scaffold with proper mechanical property implanted. DHI of the NP Injury disc or implanted with HA scaffold is decreased to ~70% of initial height at 1 week, which cannot provide effective support. As far as we know, it is difficult for previous tissue engineering NP scaffolds to maintain DHI with untreated AF puncture hole, let alone the condition that the disc suffers daily stress load after surgery.^{18, 19} PGD NP scaffold in our study maintains normally DHI under daily disc load and untreated AF puncture hole after implantation, which facilitates the surgical treatment and rehabilitation of patients.

7. It is demonstrated that curling of scaffold occurs in water in vitro at 37 degree (supplementary video). However, in the in vivo study the NP was sucked out and hence there is not enough water at the site. They should explain if the PGD rod can curl in the absence or minimal presence of moisture. Curling of scaffold in air at 37C should be demonstrated.

Response:

Thanks for the suggestions. Supplementary Video 2 is provided to reflect shape transformation

of PGD scaffold in 37 °C air. Shape transition temperature of PGD used to fabricate PGD NP scaffolds was optimized previously to achieve shape fixation at room temperature and shape transition triggered at 37 °C based on adjusting thermodynamic properties of the semi-crystalline polyester. As shown in Supplementary Video 1 and Video 2, PGD NP scaffold performs a very sensitive shape memory ability, allowing for quick deformation in both air and water condition at 37 °C.

8. Pain markers should also be tested post-implantation to confirm the functional repair.

Response:

Thanks for the suggestion. Evaluating pain level before and after disc treatment plays an important role in measuring treatment effect in the clinic. In addition to the patient description and scale evaluation from patients, medical image analysis is objective and essential for analyzing pain indicators of disc degeneration in the clinic, which is suitable for animal experiments. We analyze the possible causes of postoperative pain in animal models from the osteophytes incidence and compressive myelopathy incidence at 8 and 16 weeks after surgery according to the pathological analysis and the MRI images respectively.

Typical HE-stained disc sections with osteophytes were shown in Supplementary Fig. 3, and osteophytes incidence among each group was listed in Supplementary Table 1. We have added the description in revised manuscript at Page 10 line 201: “The results showed that osteophytes incidence of the disc in NP Injury group, HA group and HA+SDF-1 α group were about 75%. Osteophytes occurred in only 1 of the 8 discs in both PGD group and PGD+SDF-1 α groups, among which osteophytes in the disc of PGD+SDF-1 α group was milder than the other groups (Supplementary Fig. 3). The discs in NP Injury group, HA group and HA+SDF-1 α group were all lack of mechanical support, which led to lose disc height and generate osteophytic after implantation. Severe disc height losing and osteophytic might compress the peripheral nerves and caused pain. It was worth noting that osteophytes incidence in HA+SDF-1 α group at 8 weeks was significantly higher than that in other groups without SDF-1 α , which might be

related to its inappropriate stem cells recruited in the outer AF”. Previous studies have found that extruded stem cells lead to osteophytes formation near the AF wound.^{8,20} As the description added in revised manuscript at Page 11 line 237: “MRI images of the spine were used to analyze compressive myelopathy that might be caused by disc dislocation, spinal stenosis, or spinal fracture. Only one case of mild nerve compression was found in HA+SDF-1 group, and it did not cause hind limb paralysis of the rabbit. Severe compressive myelopathy causes pain and hind limb paralysis, and this phenomenon did not occur in all the experimental rabbits in this study”.

Supplementary Figure 3. Pathological morphology of the disc with osteophyte in each experimental group. HE images were taken from typical disc samples collected at 16 weeks after

implantation, and AF / osteophyte were magnified to visualize their internal cellular distribution. Black dotted box indicated areas of osteophyte generated. Red dotted lines represented AF outline of the disc. Black arrows indicated chondrocytes in AF / osteophyte.

9. Magnified and clear images of AF and NP cellularity and morphology should be included in the manuscript to correlate the presented data.

Response:

Thanks for the suggestion. Fig. 5 and Fig. 6 are modified to present variation of disc from cellular and tissue levels respectively. Distribution of cells and microvascular at the NP-EP boundary are shown in Fig. 5a. Cell morphology and stem cell composition in NP are shown in Fig. 5b. AF organization is shown by high-contrast images in Fig. 6a. Cell morphology of AF is shown in Supplementary Fig. 3.

10. The developed material is non-porous in nature and the sdf1 α will be presently mostly on the surface of the device. It means most of the stem cells will be recruited on the surface of the implant material. Then how they reach to the bulk of the material after implantation as shown in H&E and other images. Also, what type of stem cells will be recruited and how? IVD is mostly avascular tissue and thus may not be able to recruit cells from blood. Does it recruit cells from the blood supply in the endplates and how? It should be discussed with reference in the discussion.

Response:

Thanks for the comments. PGD scaffold in this study is used as a biofactor carrier to minimally invasive deliver SDF-1 α and provide mechanical support in NP cavity. Unlike tissue engineering scaffolds carrying stem cells, promoting cell growth, migration and proliferation on the scaffold is not the primary task for PGD scaffolds. we believe that the recruited cells can live in tissue fluid and extracellular matrix of NP cavity, as shown in Figure 5b. PGD scaffolds, as the mechanical support, maintain the disc height after implantation and provide physical space in NP for tissue regeneration, which is necessary for regenerate the degenerated disc. On

the contrary, HA scaffolds cannot provide enough mechanical support after implantation, which lead to SDF-1 α extrusion along the AF puncture hole and inward AF bulging into NP cavity.

Recent studies have shown that avascular tissues, such as meniscus, cartilage articular, AF, and NP etc., are able to achieve tissue regeneration by recruiting autologous stem cells.^{21, 22, 23, 24} The autologous stem cells are believed to be derived from adjacent tissues, bone marrow, and surrounding microvasculature.^{5, 25} The type of stem cells recruited varies depending on the chemokine used, and description in revised manuscript at Page 14 line 299: “SDF-1 α recruited cells with CXC chemokine receptor 4 such as bone marrow derived MSCs to participate in tissue regeneration”.¹⁷ Most of MSCs can be specifically stained by CD90, CD105, and CD166, etc.²⁶ As description in revised manuscript at Page 16 line 311: “CD90 and CD166 were selected as specific biomarkers in our study to characterize the proportion of MSCs in NP. The MSCs proportion in NP of PGD+SDF-1 α group at 8 weeks was 36.85%, which was significantly higher than that of intact group (29.32%), NP Injury group (17.84%), PGD group (15.96%), HA group (23.53%), and HA+SDF-1 α group (28.43%). CD90 and CD166 co-positive cells were also highly expressed in the microvascular and bone marrow of EP, and the number of microvascular in PGD+SDF-1 α group was significantly higher than that in PGD group and NP Injury group at 8 weeks. This phenomenon was consistent with previous studies that SDF-1 α promoted micro-angiogenesis and facilitated stem cells migration from the microvascular.⁵ The correlation between the MSCs proportion in NP and the microvascular in EP made us believe that stem cells in NP of PGD+SDF-1 α group mostly migrated from EP and its microvascular.”

Reviewer #2 (Remarks to the Author):

In this paper, the shape memory polymer poly(glycerol-dodecanoate) scaffold with chemokine stromal cell-derived factor-1 α was applied in treating intervertebral disc degeneration. And the major conclusion is that this treatment improves the disc height and recruit autologous stem cells. The data is not sufficient for the conclusion. In addition, the novelty of PGD scaffold with SDF-1 is limited. Here are some questions:

1. Why the PGD scaffold with SDF-1 has significant regenerative effects on degenerated NP, just by providing mechanical property and recruiting autologous stem cells? How many MSCs were recruited to the NP and survived after 16 weeks? More evidences should be provided to make readers convinced.

Response:

Thanks for the reviewer's suggestions, which are very important to improve our manuscript and clarify the key points in this study. As we know, cell therapy is widely researched in degenerated discs treatment in the clinic.²⁶ Except for delivering stem cells, previous studies have proved that delivering chemokine, including SDF-1 α , in NP significantly promotes the migration of autologous MSCs and restores degenerated disc function in the non-stress load disc model.^{17, 24, 27} Generally, HA was used to delivering chemokine to explore the regeneration, but it lacked effective mechanical support then resulted in extrusion of scaffold and its delivered biofactor after implantation in previous studies.^{8, 20} HA scaffolds was not the best choice in the disc treatment with AF wounds or bearing daily load after implantation.¹⁶ PGD has better mechanical properties with similar NP and shape memory ability, which satisfied with minimal invasive implantation. As above mentioned, we proposed an innovative minimally invasive biodegradable PGD scaffold to achieve chemokines delivery and effective mechanical support in NP. The PGD+ SDF-1 α scaffold is proven to improve disc degeneration from the aspect of tissue/cellular morphology, disc height, T2 MRI signal intensity of NP, disc mechanical property. In contrast, the disc of NP Injury group without scaffold implantation shows a continuous degeneration over a period of 16 weeks; the disc of PGD group without SDF-1 α delivering also shows degeneration after 8 weeks when the scaffold degraded in NP; the disc of HA and HA+ SDF-1 α group without mechanical support loses its contents in NP after implantation under daily disc load, as shown in Fig. 3f and supplementary Video 3.

In addition, previous study injects fluorescence-labeled MSCs to observe them and quantify their survival rate after implantation.⁸ Autologous MSCs recruited by SDF-1 α in our study are difficult to evaluate recruitment effect and cell survival rate through the above-mentioned

method. We suppose to calculate MSC proportion in NP Cells at 8 weeks and 16 weeks, together with comparing the proportion and pathological indexes with the intact group, NP Injury group, and other NP scaffold group, to reflect recruitment efficacy of MSCs and NP degeneration levels in our PGD+ SDF-1 α scaffold. As description in revised manuscript at Page 16 line 331:“ MSCs could be specifically stained by CD90, CD105, CD106, and CD166, etc. as previous mentioned.⁹ CD90 and CD166 were selected as specific biomarker in this study to characterize the MSCs proportion in NP. The MSCs proportion in NP of PGD+SDF-1 α group at 8 weeks was 36.85%, which was significantly higher than that of intact group (29.32%), NP Injury group (17.84%), PGD group (15.96%), HA group (23.53%), and HA+ SDF-1 α group (28.43%). This phenomenon represented that the SDF-1 α recruited MSCs in NP could only be achieved under chemokines itself exist and mechanical support from the scaffold simultaneously, and the lack of mechanical support would lead to inward AF bulging into NP and the contents extrusion (Fig. 6a HA and HA+ SDF-1 α group). The MSCs proportion in NP of PGD+SDF-1 α group at 16 weeks was 28.46%, which was as much as that of the intact disc (29.32%), PGD group (23.76%), HA+SDF-1 α group (27.12%), and higher than that of the NP Injury group and HA group. This phenomenon represented that MSCs recruitment efficacy in NP of PGD+SDF-1 α group was decreased at 16 weeks. The disc degeneration level of PGD+SDF-1 α group was improved and came to an equilibrium state from the aspect of histological evaluation, disc height, T2 MRI signal intensity of NP, disc mechanical property (Fig. 6 and Fig.7). In contrast, the disc degeneration level of NP Injury group, PGD group, HA group, and HA+SDF-1 α group got in worse as time goes by, because of no effective MSCs recruitment in NP during the 16 weeks”. For cell survival rate in NP, the modified Fig. 5b and supplementary Fig 2b showed the cell morphology with high magnification HE images at 8 weeks and 16 weeks, respectively. No obvious apoptotic cells were observed in PGD+SDF-1 α group, and the results was also reflected in histological evaluation as shown in Supplementary Table 3.

Histological scores		AF Cellularity	AF Morphology	AF-NP Border	NP Cellularity	NP Morphology	Total Scores
		(1-3)	(1-3)	(1-3)	(1-3)	(1-3)	(1-15)
Intact		1.00±0.00	1.00±0.00	1.25±0.50	1.50±0.58	1.00±0.00	5.75±0.96
NP Injury	8 W	2.00±0.00	1.50±0.58	2.50±0.58	2.00±0.00	1.75±0.50	9.75±1.26

	16 W	2.25±0.50	2.75±0.50	2.75±0.50	2.00±0.00	2.75±0.50	12.50±1.29
PGD	8 W	1.25±0.50	2.25±0.50	2.00±0.00	2.00±0.00	1.75±0.50	9.25±0.96
	16 W	1.50±1.00	2.50±0.58	2.00±0.00	1.25±0.50	1.75±0.50	9.00±0.82
PGD+SDF-1 α	8 W	1.25±0.50	2.25±0.50	1.75±0.50	1.50±0.58	1.75±0.50	8.25±0.51
	16 W	1.25±0.50	2.00±0.82	2.00±0.00	1.25±0.50	1.75±0.50	7.50±1.29
HA	8 W	1.50±1.00	2.25±0.50	2.00±0.82	3.00±0.00	2.00±0.82	10.75±1.71
	16 W	2.00±0.82	2.50±0.58	2.25±0.50	3.00±0.00	2.25±0.50	12.00±1.00
HA+SDF-1 α	8 W	2.25±0.96	2.00±0.82	2.25±0.50	2.00±0.00	2.25±0.50	10.75±1.71
	16 W	1.25±0.50	1.75±0.96	2.50±0.58	2.75±0.50	2.25±0.50	10.50±1.00

Total Scores (Slight inflammation: 5-8; Medium inflammation: 8-12; Severe inflammation: 12-15)

Supplementary Table 3. Histological scores of each experimental group from AF cellularity, AF Morphology, AF-NP border, NP cellularity, and NP morphology parts.

2. Why CD90+&CD166+ cells were decreased after 16W compared with 8W in the PGD+SDF-1 group, but the regenerative effects were increased?

Response:

Thanks for the comments. Microenvironment of the degenerated tissue can promote the inside cells secreting biofactors, such as chemokines, to recruit MSCs in the surrounding tissues and peripheral vascular to realize tissue regeneration.^{14, 27} Current studies believe that tissue repair function of MSC is closely related to its secretion and extracellular vesicles mediated inhibitory effect of inflammation and apoptosis, and cell differentiation effect.¹³ Avascular tissues like NP cannot repair itself after degeneration due to the limited stem cells recruit ability.^{21, 28} Chemokine SDF-1 α is used in our study to recruit autologous MSCs into NP and restore degenerated disc function in rabbits lumbar model.

As above mentioned, CD90 and CD166 are selected as specific biomarkers in our study to characterize the MSCs proportion in NP.²⁶ As description in revised manuscript at Page 16 line 331, A higher CD90 & CD166 co-positive cells proportion in NP of PGD+SDF-1 α group at 8 weeks means a better SDF-1 α mediated MSCs recruitment in NP than other groups, following a better tissue repair function from recruited MSC. The results of repair function can be confirmed by histological evaluation, disc height variation, NP T2 MRI signal intensity variation, and mechanical property test in our study (Fig. 6 and Fig. 7). The disc of PGD+SDF-

1 α group does experienced better function regeneration than other groups during 16 weeks implantation. As description in revised manuscript at Page 16 line 345: As for the decreased CD90 & CD166 co-positive cells at 16 weeks in PGD+SDF-1 α group, it should note that the co-positive cells are as much as the intact disc, which means the MSCs recruitment function is decreased. The disc degeneration level of PGD+SDF-1 α group comes to an equilibrium state that is close to the intact disc at 16 weeks (Fig. 6 and Fig. 7). In contrast, the disc degeneration level of NP Injury group, PGD group, HA group, and HA+SDF-1 α group get in worse at 16 weeks, because of no effective MSCs recruitment in NP during the 16 weeks.

3. In Fig.6, the NP was obvious in the intact group, but the area of NP could not be distinguished with AF after 16W in each group. There is a significant difference between healthy NP and PGD+SDF-1 treated NP.

Response:

Thanks for the suggestion. NP-AF boundary morphology in Fig. 6 is modified with high contrast black and white images to reflect AF organization clearly at 8 weeks and 16 weeks among the Intact, NP Injury, and four NP scaffold groups. Also, the red dashed curves are used to mark AF organization in Fig. 6a to allow a more obvious visibility of morphological changes at different disc degeneration degrees. The NP Injury group is added to observe disc degradation process with injured NP and relative intact AF. The intact disc is found to have a clear NP-AF boundary and well-organized collagen lamellae without ruptured or serpentine fibers; the disc in NP Injury group showed a gradual severe degeneration with time; the disc in PGD group showed normal at 8 weeks due to the support capability provided by PGD scaffold, and inward bulging AF fibers are generated at 16 weeks for gradual degradation of PGD scaffold; PGD+SDF-1 α group maintained the basic morphologies of AF and NP at 8 and 16 weeks without obvious degeneration; inward bulging AF in HA group is observed at 8 weeks due to the lack of mechanical support and hydrogel extrusion, osteophytes formed near AF puncture site at 16 weeks; AF in HA+SDF-1 α group is similar to HA group, and osteophytes formed near AF puncture site as early as 8 weeks.

PGD+SDF-1 α group does show slight degenerative features at 8 and 16 weeks compared with the intact group (Fig. 6a), including serpentine fibers in AF and misty NP-AF boundary, which is reflected in the histopathology evaluation as shown in Supplementary Table 3. However, the degeneration process of the disc in PGD+SDF-1 α group is inhibited, which showed no significant differences with the intact disc from NP cell morphology, disc height, T2 MRI signal, and mechanical properties at 16 weeks after implantation (Fig. 5, and Fig. 7). In contrast, the degeneration process of the disc in NP Injury group is severe gradually within 16 weeks; HA and HA+SDF-1 groups experiences disc height loses at the very beginning because of lacking effective support; PGD group shows gradual disc degeneration from 8 weeks to 16 weeks due to the scaffold degradation. As far as we know, it is difficult for previous tissue engineering NP scaffolds to maintain disc height with untreated AF puncture hole, let alone the condition that the disc suffers daily stress load after surgery.^{18, 19} PGD NP scaffold in our study maintains clinical indexes and mechanical property close to the intact disc at 16 weeks base on the premise of both untreated AF puncture hole and daily disc load after implantation, which is facilitate the surgical treatment and rehabilitation of patients.

Reviewer #3 (Remarks to the Author):

The authors present a tissue engineering study for the application of a biomaterial that can keep its shape in a “memory”. Furthermore, the material allows releasing a growth factor, in this case, SDF-1 α . This growth factor is known to recruit mesenchymal stromal cells (and other progenitor cells) and is important for the field of regenerative medicine and the field of orthopaedic research on the spine. The manuscript is on the innovation of shape-memory materials that could find their way to clinical application.

The study is supported by in-vitro biomechanical characterization of the material and by an in-vivo rabbit model for up to 9 weeks subcutaneously. Biocompatibility of the material was also tested in a subcutaneous model in Sprague-Dawley rats.

Finally, the animal model (New Zealand white rabbit) is back-upped by numerical simulations

of the PGD NP scaffold.

Abstract: The abbreviations should be already introduced in the abstract, i.e., SDF-1 α and PGD as the abstract should be independent of the main text.

Response:

Thanks for the comments. We add abbreviations to the abstract and proofread them carefully.

Page 6 line 109: please, recall for the reader the young's modulus (compressive Modulus) for a healthy non-degenerated IVD (for both, NP and AF). Possibly, also provide a reference for it.

Response:

Thanks for the comments. Young's modulus (compressive Modulus) for a healthy non-degenerated IVD for both, NP and AF are added to the manuscript, as description in revised manuscript at Page 6 line 107: "Then, a set of PGD synthesis parameters ($MR_{H/C}=1.50$, $t=72h$) closest to the compressive modulus of native NP (1.01 ± 0.43 MPa) was finally selected and verified to fabricate our NP scaffold, also the compressive modulus of AF was known as 440-750 kPa.²⁹"

Line 374: Please, specify the standard of ASTM. For what does ASTM stand for?

Response:

Thanks for the comments. The full name of ASTM is added as description in revised manuscript at Page 19 line 406, and abbreviations in the manuscript are all checked.

Line 378: Please, provide brand and city of manufacturer of modified acrylate, also specify in what way the acrylate was modified.

Response:

Thanks for the comments. The modified acrylate used in our study is a commercial product (GLH corporation, Fushun China) with fast cure and high strength abilities, which is suitable to embedded the end of vertebra for mechanical test, as shown in Fig. 7g.

Fig. 7g

Line 416: please, add scanning resolution of the μ CT and slice thickness of the scans.

Response:

Thanks for the comments. Scanning resolution of the μ CT and slice thickness of the scans are added as following in revised manuscript at Page 21 line 456: The finite element model of the vertebra-disc was developed based on Micro-CT (SkyScan1276, Bruker) images of the New Zealand rabbit lumbar segment with 20 μ m pixel resolution and 100 μ m slice thickness.

Line 492: Four parallel rabbit samples graded the histological scores for each group via HE, Masson, Alcian Blue, and Safranin O staining images. \diamond strange wording, rephrase this sentence

Response:

Thanks for the comments. The sentence is improved as following in revised manuscript at Page 25 line 537: The histological scores for each group were evaluated via HE, Masson, Alcian Blue, and Safranin O staining images from four rabbits' samples.

Line 501: Please, provide working concentrations of CD90 and CD166 primary antibodies and secondary antibodies.

Response:

Thanks for the comments. Working concentrations of CD90 and CD166 primary antibodies and secondary antibodies are added as following in revised manuscript at Page 25 line 545: The slices were blocked with 5% BSA solution, incubated with 1:10 diluted CD90 and CD166 primary antibodies (Abcam) for 12 hours at 4°C, followed by incubation with 1:200 diluted

Alexa-fluor 488 and Alexa-fluor 647-labeled secondary antibodies (ThermoFisher) for 1 hour at room temperature.

Lines 504 and the following: describe your negative and positive controls for the immune staining.

Response:

Thanks for the comments. The description is described in the revised manuscript at Page 26 line 550: “For each immune stained section, the cells in the bone marrow and growth plate of vertebra are used as positive controls for CD90 & CD166 co-positive cells, while osteocyte cells in vertebra are used as negative controls, as shown in Fig. R3-1.”

Figure R3-1. Positive (+) and negative (-) controls for the immune staining

Figure 6 b and c: with N = 4 data points per group the bar plots should be replaced with single data point plots and overlaying with means \pm SD or SEM.

Figure 7 k and l: also, here replace bar plots with single data plots and overlay means \pm SEM.

Response:

Thanks for the comments. Single data of each group is added in Fig. 6 b, c and Fig. 7 k, l, and their means \pm SD are also labeled behind.

Fig. 6 b,c

Fig. 7 k,l

Discussion

The authors should include a discussion on the limitations of the rat and the rabbit model for IVD research. These animal groups are very different from the human spine in many aspects. Most importantly, these animals are much more regenerative as the IVDs contain a high number of notochordal cells and the NP is much more jelly-like than in the human situation.

Response:

Thanks for the comments. Limitations of the rat and rabbit model used in this paper are discussed in revised manuscript at Page 17 line 366, as following: There were still limitations in our study. New Zealand rabbits and SD rats used in our study was very different from the human spine in many aspects. These animals were much more regenerative as the disc contain a high number of notochordal cells and the NP was much more jelly-like than in the human situation. Further experiments in large experimental animals or human were needed to demonstrate the effectiveness of the PGD NP scaffold described herein.

The figures and the legends in general are fine. However, following the trend of showing all the data points in the graphs I recommend the authors to switch from bar plots to single data graphs.

Response:

Tanks for the comments. Single data of each group in the bar plots is added, including: Fig. 4f, Fig. 5c,d,e,f, Fig. 7b,c,d,e,f,g.

Fig. 4 f

Fig. 5 c,d,e,f

Fig. 7 b,c,d,e,f,g

The authors do show very nice histology using paraffin. For paraffin, the tissue needed to be decalcified prior to embedding. Are the authors not afraid that this step would generate artefacts as also proteoglycans are washed out during this step? Would not hard sections be the better option?

Response:

Thanks for the comments. Hard sections of the disc are considered in our pre-experiments, but it is not as good as paraffin sections we believe after weighing the advantages and disadvantages. Because it is necessary in our study to take multiple sections from the same part of the disc for four kinds of pathological staining and immunofluorescence staining, which requires multiple sections close to each other from the small sized rabbit disc, while hard sections cannot meet above conditions well. Reviewing previous related studies, paraffin sections have usually been used to analyze proteoglycans content in the disc.^{30, 31, 32} Thus, we choose to use paraffin sections in our study. The artifacts and proteoglycan elution caused by decalcification process of paraffin embedding mentioned by your comments are also very important, and we will pay full attention to them in our future study.

Fig.6 should add in the caption that these are paraffin sections.

Response:

Thanks for the comments. Figure legend of Fig. 6 is changed, as following: Histological paraffin sections images and evaluation of disc degeneration after 8, 16 weeks implantation.

References

1. Endres M, *et al.* Intervertebral disc regeneration after implantation of a cell-free bioresorbable implant in a rabbit disc degeneration model. *Biomaterials* **31**, 5836-5841 (2010).
2. Endres M, *et al.* Augmentation and repair tissue formation of the nucleus pulposus after partial nucleotomy in a rabbit model. *Tissue & Cell* **46**, 505-513 (2014).
3. Tsujimoto T, *et al.* An acellular bioresorbable ultra-purified alginate gel promotes intervertebral disc repair: A preclinical proof-of-concept study. *Ebiomedicine* **37**, 521-534 (2018).

4. Sive JI, Baird P, Jeziorski M, Watkins A, Hoyland JA, Freemont AJ. Expression of chondrocyte markers by cells of normal and degenerate intervertebral discs. *J Clin Pathol-Mol Pa* **55**, 91-97 (2002).
5. Zhang HX, Wang P, Zhang X, Zhao WR, Ren HL, Hu ZM. SDF1/CXCR4 axis facilitates the angiogenesis via activating the PI3K/AKT pathway in degenerated discs. *Mol Med Rep* **22**, 4163-4172 (2020).
6. Xu HM, Hu F, Wang XY, Tong SL. Relationship Between Apoptosis of Endplate Microvasculature and Degeneration of the Intervertebral Disk. *World Neurosurgery* **125**, E392-E397 (2019).
7. Ashinsky BG, *et al.* Intervertebral Disc Degeneration Is Associated With Aberrant Endplate Remodeling and Reduced Small Molecule Transport. *Journal of Bone and Mineral Research* **35**, 1572-1581 (2020).
8. Vadala G, Sowa G, Hubert M, Gilbertson LG, Denaro V, Kang JD. Mesenchymal stem cells injection in degenerated intervertebral disc: cell leakage may induce osteophyte formation. *J Tissue Eng Regen Med* **6**, 348-355 (2012).
9. Binch ALA, Fitzgerald JC, Growney EA, Barry F. Cell-based strategies for IVD repair: clinical progress and translational obstacles. *Nat Rev Rheumatol* **17**, 158-175 (2021).
10. DiStefano TJ, Vaso K, Danias G, Chionuma HN, Weiser JR, Iatridis JC. Extracellular Vesicles as an Emerging Treatment Option for Intervertebral Disc Degeneration: Therapeutic Potential, Translational Pathways, and Regulatory Considerations. *Adv Healthcare Mater* **11**, 2100596 (2022).
11. Guerrero JP, Häckel S, Croft AS, Hoppe S, Albers C, Gantenbein B. The nucleus pulposus microenvironment in the intervertebral disc: the fountain of youth? *European cells & materials eCM* **41**, 707-738 (2021).
12. Dou YM, Sun X, Ma XL, Zhao X, Yang Q. Intervertebral Disk Degeneration: The Microenvironment and Tissue Engineering Strategies. *Front Bioeng Biotechnol* **9**, 592118 (2021).
13. Peng YZ, *et al.* Decellularized Disc Hydrogels for hBMSCs tissue-specific differentiation and tissue regeneration. *Bioact Mater* **6**, 3541-3556 (2021).
14. Sakai D, Grad S. Advancing the cellular and molecular therapy for intervertebral disc disease. *Adv Drug Deliver Rev* **84**, 159-171 (2015).
15. Ko IK, Lee SJ, Atala A, Yoo JJ. In situ tissue regeneration through host stem cell recruitment.

Exp Mol Med **45**, e57 (2013).

16. D'Este M, Eglin D, Alini M. Lessons to be learned and future directions for intervertebral disc biomaterials. *Acta Biomater* **78**, 13-22 (2018).
17. Zhang H, Yu S, Zhao X, Mao Z, Gao C. Stromal cell-derived factor-1 α -encapsulated albumin/heparin nanoparticles for induced stem cell migration and intervertebral disc regeneration in vivo. *Acta Biomater* **72**, 217-227 (2018).
18. Sloan Jr SR, *et al.* Combined nucleus pulposus augmentation and annulus fibrosus repair prevents acute intervertebral disc degeneration after discectomy. *Sci Transl Med* **12**, eaay2380 (2020).
19. Bowles RD, Setton LA. Biomaterials for intervertebral disc regeneration and repair. *Biomaterials* **129**, 54-67 (2017).
20. Willems N, *et al.* Intradiscal application of rhBMP-7 does not induce regeneration in a canine model of spontaneous intervertebral disc degeneration. *Arthritis Res Ther* **17**, 1-15 (2015).
21. Lee CH, Cook JL, Mendelson A, Muioli EK, Yao H, Mao JJ. Regeneration of the articular surface of the rabbit synovial joint by cell homing: a proof of concept study. *Lancet* **376**, 440-448 (2010).
22. Lee CH, Rodeo SA, Fortier LA, Lu CY, Erisken C, Mao JJ. Protein-releasing polymeric scaffolds induce fibrochondrocytic differentiation of endogenous cells for knee meniscus regeneration in sheep. *Sci Transl Med* **6**, 266ra171 (2014).
23. Liu C, Jin ZX, Ge X, Zhang Y, Xu HG. Decellularized Annulus Fibrosus Matrix/Chitosan Hybrid Hydrogels with Basic Fibroblast Growth Factor for Annulus Fibrosus Tissue Engineering. *Tissue Eng Part A* **25**, 1605-1613 (2019).
24. Pereira CL, *et al.* The effect of hyaluronan-based delivery of stromal cell-derived factor-1 on the recruitment of MSCs in degenerating intervertebral discs. *Biomaterials* **35**, 8144-8153 (2014).
25. Fournier DE, Kiser PK, Shoemaker JK, Battie MC, Seguin CA. Vascularization of the human intervertebral disc: A scoping review. *Jor Spine* **3**, e1123 (2020).
26. Lyu FJ, Cheung KM, Zheng ZM, Wang H, Sakai D, Leung VY. IVD progenitor cells: a new horizon for understanding disc homeostasis and repair. *Nat Rev Rheumatol* **15**, 102-112 (2019).
27. Ying J, Han Z, Pei S, Su L, Ruan D. Effects of stromal cell-derived factor-1 α secreted in degenerative intervertebral disc on activation and recruitment of nucleus pulposus-derived stem cells. *Stem Cells Int* **2019**, 9147835 (2019).

28. Leite Pereira C, *et al.* Stromal cell derived factor-1-mediated migration of mesenchymal stem cells enhances collagen type II expression in intervertebral disc. *Tissue Eng Part A* **24**, 1818-1830 (2018).
29. Nerurkar NL, Elliott DM, Mauck RL. Mechanical design criteria for intervertebral disc tissue engineering. *Journal of Biomechanics* **43**, 1017-1030 (2010).
30. Feng GJ, *et al.* Injectable nanofibrous spongy microspheres for NR4A1 plasmid DNA transfection to reverse fibrotic degeneration and support disc regeneration. *Biomaterials* **131**, 86-97 (2017).
31. Leung VYL, *et al.* Mesenchymal Stem Cells Reduce Intervertebral Disc Fibrosis and Facilitate Repair. *Stem Cells* **32**, 2164-2177 (2014).
32. Sloan SR, *et al.* Combined nucleus pulposus augmentation and annulus fibrosus repair prevents acute intervertebral disc degeneration after discectomy. *Sci Transl Med* **12**, eaay2380 (2020).

REVIEWER COMMENTS

Reviewer #1 (Remarks to the Author):

1. The previous comment “Injury group is missing in fig. 5, 6 and 7, How the injury was created when the outer AF remain intact as shown in figure 6. Also, inserting .2 mm injury is big to appear on H&E data and model should be validated before implantation” has not been addressed completely. It is important to see the injury in AF area to confirm the procedure. In the mentioned procedure needle puncture was used which goes through the AF region and destabilize the NP area. The reviewer wanted to see the injury in the AF region also which is unavoidable during the procedure and may trigger IVD degradation.
2. The MRI and X-ray images to confirm scaffold is inconclusive. Direct staining method or tracker should be used.
3. Recruitment of MSCs are shown but still no study has been conducted to demonstrate its differentiation in to NP cells and restoring the cellular environment which limits its translational to clinic.
4. In figure 6, organization of AF is after 16w of implantation in PGD group is still not convincing.

Reviewer #3 (Remarks to the Author):

The authors have addressed well all of the reviewer's comments. I do not have additional remarks.

REVIEWER COMMENTS

Reviewer #1 (Remarks to the Author):

1. The previous comment “Injury group is missing in fig. 5, 6 and 7, How the injury was created when the outer AF remain intact as shown in figure 6. Also, inserting .2 mm injury is big to appear on H&E data and model should be validated before implantation” has not been addressed completely. It is important to see the injury in AF area to confirm the procedure. In the mentioned procedure needle puncture was used which goes through the AF region and destabilize the NP area. The reviewer wanted to see the injury in the AF region also which is unavoidable during the procedure and may trigger IVD degradation.

Response:

Thanks for the reviewer’s comments. We totally agreed that AF injury is unavoidable both in clinical discectomy or tissue engineering approaches, which may trigger IVD degeneration and is important for the treatment of degenerated IVD. We clarified puncture site of hollow needle in Figure R1-1, which located at the oblique posterior of lateral disc. Pathological sections through the puncture site are selected to clarify AF injury of each group at 8 and 16 weeks after implanted in Supplementary Figure 3. Yellow dotted box in the images of the figure represents the puncture injury in AF, and the magnification of the injury site is marked by the black-white images for a detailed observation. The following content is added to the revised manuscript at page 10 line 7. The AF puncture injury of each group do not heal completely, and it is still clear at 16 weeks after implantation. The disc of NP injury group cannot maintain the osmotic pressure in NP due to the AF injury, leading to poor mechanical support and disc height loss at 8 weeks after implantation. The abnormal organization of AF in NP injury group are shown in the black-white images of Supplementary Figure 3. Serious degeneration of the disc in NP injury group are shown at 16 weeks after implantation for the inner convex of AF into NP cavity. Same phenomenon also presented in HA and HA+SDF-1 α group, indicating that filling NP with HA alone is not sufficient to maintain physical function of the disc with AF injury.

Figure R1-1 The puncture site of hollow needle during surgery. a) The puncture site was in the oblique posterior of lateral disc; b) Approximate paraffin section shown in Figure 6 and Supplementary Figure 3.

As Sloan and Wipplinger et al. concluded in the published in the journal of “Sci Transl Med” (Sci Transl Med, 2020, 12(534): eaay2380), combined NP filling and AF repair is necessary to maintain the physical function of degenerated discs.¹ We think it was a good idea. In this study, we just try to realize the same effect with the brand-new approach with NP scaffold itself. PGD NP scaffold is adopted in our study to present compressive modulus similar to that of native NP at restricted condition, providing mechanical support after implantation to withstand daily activities of the disc with AF injury. In the previous studies, researches are hardly to fulfill the both function in one scaffold.² With the shape deformation ability in NP cavity, PGD NP scaffold is able to deliver through a minimal invasive needle just like the hydrogel scaffold, and provide sufficient mechanical support like the scaffold made by tough polymer. The increased cross-section area of the shape deformed PGD NP scaffold after implantation makes it stable in the NP cavity. As shown in Figure 6 and Supplementary Figure 3, NP cavity of PGD and PGD+SDF-1 α group maintains well at 8 weeks, which provides necessary physical space for the recruited MSCs and improves the disc function regeneration. The disc morphology of PGD+SDF-1 α group shows better than that of the NP injury and other three scaffold groups at 16 weeks with the help of MSCs’ regenerate ability. AF injury of PGD+SDF-1 α group also improves compared with the NP injury and other three scaffold groups, indicating an inhibited disc degeneration and good regeneration ability; AF organization near the injury site in NP injury group shows internal convex into NP cavity due to the loss of NP in disc; hydrogel-injected HA and HA+SDF-1 α groups shows both internal convex and external convex AF. Osteophytes are also

formed outside the puncture injury, which was related to extrusion of the contents in NP after surgery.

Supplementary Figure 3 Safranin O and Masson-stained images of native intact disc, NP injury disc, and disc implanted with four NP scaffolds in the coronal plane at 8 and 16 weeks. Yellow dotted box in the images represented the injury in the AF region. Red dotted lines represented AF outline of the disc.

2. The MRI and X-ray images to confirm scaffold is inconclusive. Direct staining method or tracker should be used.

Response:

Thanks for the reviewer's comment, which is very important to improve the manuscript. We totally agreed that generally, direct staining method and tracker would provide good indication of the medical implants during its implanting. But it needs to modify the scaffold material using radiographic or fluorescent tracer firstly. As we know, there is no safe and good modification methods applied in the NP tissue engineering scaffold. So, it needs to be confirmed that the effect of the modification on the degradable NP scaffold and the disc itself before surgery, including the effect of mixed tracer on the fabrication of PGD scaffolds, the mechanical properties of PGD polymer, and bioactivity of the chemokines loaded on the NP scaffold. In addition, the effects of radiographic or fluorescent tracers on tissue repair in avascular intervertebral discs. We are focusing on the effect and explore the most effective methods in the following study. In this study, we just observed the MRI, X-ray images, and pathological sections to followed the position of NP scaffolds since we hypothesized it would be stay in the same position during degradation. It would be valuable to explored the effect of mixed tracer and find a best one, which was meaningful to the related research of medical implants. The above mentioned are clarified in the revised manuscript at page 18 line 12.

3. Recruitment of MSCs are shown but still no study has been conducted to demonstrate its differentiation in to NP cells and restoring the cellular environment which limits its translational to clinic.

Response:

Thanks for the reviewer's valuable suggestions, which is a very important question especially in the field of tissue engineering. It would guide us to understand the limitations of our study and proceed our follow-up studies. In this study, we focused on

providing a novel idea and design to achieve minimally invasive delivery and mechanical support of NP tissue with the help of PGD polymer, then explored the possibility of NP regeneration through the PGD scaffold loaded chemokines. We have explored the effect of MSCs recruitment ability and NP function regeneration of the PGD NP scaffold indirectly. A direct method to demonstrate cell differentiation of recruited MSCs is a key point in the following study. As described by Binch et al. in the published review article³, the regenerate mechanisms of the delivered/recruited MSCs in disc remains a matter with different options, although a lot of clinical studies have delivered MSCs to the degenerated disc of human and got positive effects. Differentiation of MSCs into NP cells was one of the regeneration factors. Cytokines and vesicles secreted by MSCs also play an important role in regulating the cellular environment in NP according to related studies.^{4, 5} We would focus on the corresponding *in vitro* or *ex vivo* experiments and then processing gene expression analysis of the recruited MSCs in NP in the following study. The above-mentioned limitations are also clarified in the revised manuscript at page 18 line 9.

4. In figure 6, organization of AF is after 16w of implantation in PGD group is still not convincing.

Response:

Thanks for the reviewer's comment for the figure 6. For a better understanding, we add the images as shown in Figure R1-1, middle part of the disc is selected to section and stain for pathologic analysis in Figure 6 for presenting both PGD scaffold and disc morphology. Supplementary Figure 3 was also added to show the punctured AF at 8 and 16 weeks after surgery. Yellow dotted box in the figure highlights the location of AF injury. The following content is added in the results of revised manuscript at page 10 line 7. Organization of AF in NP injury, HA, and HA+SDF-1 α groups shows typical degenerated characteristics. Thereinto, AF organization near the injury site in NP injury group shows internal convex into NP cavity due to the loss of NP in disc; hydrogel-injected HA and HA+SDF-1 α groups shows both internal convex and external convex

AF. Osteophytes are also formed outside the puncture injury, which is related to extrusion of the contents in NP after surgery. AF organization near the injury site in PGD+SDF-1 α shows same with the intact group at 16 weeks, indicating a good mechanical support provided from the PGD scaffold and tissue regeneration from recruited MSCs play an important role during the period.

Reviewer #3 (Remarks to the Author):

The authors have addressed well all of the reviewer's comments. I do not have additional remarks.

References

1. Sloan Jr SR, *et al.* Combined nucleus pulposus augmentation and annulus fibrosus repair prevents acute intervertebral disc degeneration after discectomy. *Sci Transl Med* **12**, eaay2380 (2020).
2. Li C, *et al.* Advances and prospects in biomaterials for intervertebral disk regeneration. *Front Bioeng Biotechnol* **9**, 766087 (2021).
3. Binch ALA, Fitzgerald JC, Growney EA, Barry F. Cell-based strategies for IVD repair: clinical progress and translational obstacles. *Nat Rev Rheumatol* **17**, 158-175 (2021).
4. DiStefano TJ, Vaso K, Danias G, Chionuma HN, Weiser JR, Iatridis JC. Extracellular vesicles as an emerging treatment option for intervertebral disc degeneration: therapeutic potential, translational pathways, and regulatory considerations. *Adv Healthcare Mater* **11**, 2100596 (2022).
5. Dou Y, Sun X, Ma X, Zhao X, Yang Q. Intervertebral Disk Degeneration: The Microenvironment and Tissue Engineering Strategies. *Front Bioeng Biotechnol* **9**, 592118 (2021).

REVIEWERS' COMMENTS

Reviewer #1 (Remarks to the Author):

Authors have satisfactorily answered all the queries with the understanding of limitation in the study. The alternate justifications have been given for the comments which are acceptable.